# Elephants Never Forget: Memorization and Learning of Tabular Data in Large Language Models

**Sebastian Bordt**
University of Tübingen, Tübingen AI Center
sebastian.bordt@uni-tuebingen.de

**Harsha Nori & Vanessa Rodrigues & Besmira Nushi & Rich Caruana**
Microsoft Research
{hnori,vanessa.rodrigues,besmira.nushi,rcaruana}@microsoft.com

## Abstract

While many have shown how Large Language Models (LLMs) can be applied to a diverse set of tasks, the critical issues of data contamination and memorization are often glossed over. In this work, we address this concern for tabular data. Specifically, we introduce a variety of different techniques to assess whether a language model has seen a tabular dataset during training. This investigation reveals that LLMs have memorized many popular tabular datasets verbatim. We then compare the few-shot learning performance of LLMs on datasets that were seen during training to the performance on datasets released after training. We find that LLMs perform better on datasets seen during training, indicating that memorization leads to overfitting. At the same time, LLMs show non-trivial performance on novel datasets and are surprisingly robust to data transformations. We then investigate the in-context statistical learning abilities of LLMs. While LLMs are significantly better than random at solving statistical classification problems, the sample efficiency of few-shot learning lags behind traditional statistical learning algorithms, especially as the dimension of the problem increases. This suggests that much of the observed few-shot performance on novel real-world datasets is due to the LLM's world knowledge. Overall, our results highlight the importance of testing whether an LLM has seen an evaluation dataset during pre-training. We release the tabmemcheck Python package to test LLMs for memorization of tabular datasets.

## 1 Introduction

Large Language Models (LLMs) exhibit remarkable performance on a diverse set of tasks (Wei et al., 2022; Bubeck et al., 2023; Liang et al., 2023). While their prowess in natural language is undeniable, performance in other applications remains an ongoing research topic (Dziri et al., 2023a; Nori et al., 2023). Recently, LLMs have increasingly been employed for diverse data modalities. This includes an increasing amount of creative applications in domains such as structured learning, tabular data, and time-series forecasting (Yin et al., 2023; Chen et al., 2023; Hegselmann et al., 2023; Jin et al., 2024).

A main question in current research on LLMs is the degree to which these models are able to extrapolate to novel tasks that are unlike what they have seen during training (Wu et al., 2023). As such, an important aspect of LLM evaluation is to know if a task has been part of the model's training set. In this work, we refer to the case where an LLM is evaluated on a dataset seen during (pre-)training as *training data contamination* (Magar & Schwartz, 2022; Jiang et al., 2024). Unfortunately, detecting whether an LLM has seen a certain text or dataset during training is rather challenging (Duan et al., 2024). This is especially true for models to which we might only have query access via an API.

| | GPT-3.5 | | | GPT-4 | | Baseline |
|---|---|---|---|---|---|---|
| **Time Series** | 2010 - 2019 | 2020 | 2022 | 2020 | 2022 | 2022 |
| U.S. Dollars to Yuan | 0.09% - 0.20% | 0.17% | 0.30% | 0.13% | 0.32% | 0.24% |
| U.S. Dollars to Euro | 0.12% - 0.43% | 0.18% | 0.58% | 0.14% | 0.59% | 0.44% |
| NASDAQ | 0.36% - 0.98% | 0.65% | 1.35% | 0.07% | 1.30% | 1.20% |
| MSCI World | 0.29% - 0.61% | 0.82% | 1.10% | 0.30% | 1.03% | 0.92% |
| Netflix | 0.96% - 1.54% | 0.68% | 1.58% | 0.25% | 1.60% | 1.47% |

Table 1: Forecasting financial time series with GPT-3.5 and GPT-4 exhibits remarkable performance differences between years prior to and after 2021 (the cutoff of the training data). We perform few-shot learning, providing the model with the value of the time series on the previous 20 days and asking it to predict the value on the next day. The table depicts the robust mean **relative error** of the prediction across different years. The second column depicts the minimum and maximum error across the ten years from 2010 to 2019. The baseline predicts the value of the past day for the current day. Details in Supplement C.

At the same time, a literature on *memorization* in LLMs has shown that language models can be prompted to repeat chunks of their training data verbatim (Carlini et al., 2019; 2021; Chang et al., 2023; Nasr et al., 2023). Research has also shown that memorization occurs if an LLM sees a text repeatedly during training (Carlini et al., 2022b; Biderman et al., 2023). Because of this, memorization can be seen as an extreme case of training data contamination where a dataset is not only seen during training but repeated within the training set so often that the LLM becomes able to consistently generate it.

In this paper, we target the issue of training data contamination when evaluating the few-shot learning performance of LLMs on tasks with tabular data – an aspect often neglected in the rapidly growing literature on applications in this domain (Dinh et al., 2022; Borisov et al., 2023; Narayan et al., 2022; Vos et al., 2022; Hegselmann et al., 2023; Wang et al., 2023; McMaster et al., 2023). Our first contribution is to develop various methods to detect whether an LLM has seen a tabular dataset during training. This includes four different tests for memorization. Our investigation reveals that GPT-3.5 and GPT-4 have memorized many popular tabular datasets verbatim (OpenAI, 2023). For example, GPT-4 can consistently generate the entire Iris and Wine datasets from the UCI machine learning repository. What is more, the memorization of tabular datasets in GPT-series models is a robust phenomenon that does not depend on the precise model version.

To gauge the effect of training data contamination on evaluations of state-of-the-art LLMs, we compare the few-shot learning performance of GPT-3.5 and GPT-4 on datasets that were seen during training to datasets that were released after training (Brown et al., 2020).[1] In our experimental design, we also vary the format in which the data is presented to the LLM. In particular, we add small amounts of noise to numerical values in the dataset. The idea is that contamination is more likely to have an effect if the LLM can "recognize" a datapoint from pre-training.

We find that LLMs perform better on datasets seen during training, indicating that memorization leads to overfitting. In addition, we find that adding small amounts of noise and other re-formatting techniques leads to an average *accuracy drop of 6 percentage points on the memorized datasets*. In contrast, the same transformations do not affect the few-shot learning performance on unseen data. To see the significant effect that training data contamination can have on few-shot learning, consider also Table 1. It shows that GPT-3.5 and GPT-4 exhibit remarkable performance differences when predicting time series data for time periods prior to and after the LLM's training data cutoff.

While we find significant evidence of training data contamination, we also find that GPT-3.5 and GPT-4 perform reasonably well on novel datasets. To better understand the LLMs' few-shot learning performance on the novel datasets, we conduct ablation studies and investigate

---

[1] We do not know the training data of the LLMs. We use the term "seen during training" when there is evidence of verbatim memorization of at least part of a dataset.

their performance as in-context statistical predictors (Garg et al., 2022). In particular, we remove the feature names and standardize the data to zero mean and constant variance. We find that GPT-3.5 and GPT-4 can still perform in-context statistical classification better than random but struggle as the dimension of the problem increases. This leads us to conclude that the few-shot learning performance on novel tabular datasets is due to the LLM's world knowledge. Interestingly, we also find that the performance of in-context statistical learning with GPT-4 scales in the number of few-shot examples, whereas it remains more flat for GPT-3.5.

The paper is organized as follows. We first present our memorization tests. We then study the few-shot learning performance on memorized and novel tabular datasets. We then study the ability of LLMs to act as statistical predictors during few-shot learning. Finally, we show that LLMs can draw random samples from datasets they have seen during training.

## 2   Datasets

In this study, we use two different kinds of tabular data. First, we use datasets that were freely available on the Internet before 2021. This includes the `Iris` , `Wine` , `Adult` and `Housing` datasets from the UCI machine learning repository (Kelley Pace & Barry, 1997). It also includes the OpenML `Diabetes` dataset and the `Titanic` dataset from Kaggle (Smith et al., 1988). We use red color to indicate these popular datasets.

Second, we use datasets that were not freely available on the Internet before 2022. This includes the `ACS Income` and `ACS Travel` datasets (Ding et al., 2021). These two datasets were derived from the 2022 American Community Survey (ACS). Notably, the `ACS Income` dataset was constructed by Ding et al. (2021) to obtain a dataset similar to the `Adult` dataset. We also use the `Spaceship Titanic` dataset, released on Kaggle in 2022, and the `FICO` dataset that has been privately held (the dataset is available for download after registration). We also introduce the `ICU` dataset, a novel small machine learning dataset where the task is to predict whether a patient is being treated in intensive or intermediate care. We derived this dataset from data in the Harvard dataverse (Goad, 2018). We use blue color to indicate the novel datasets. Additional details on the datasets can be found in Supplement A.

## 3   Testing Language Models for Memorization of Tabular Data

There are various systematic ways to test what a language model knows about a tabular dataset. For example, we can test whether the LLM can list a dataset's feature names or tell the possible values of categorical features in the data. It is also possible to heuristically assess whether the LLM has learned different aspects of the data distribution. In this Section, we focus on tests for verbatim memorization (Carlini et al., 2019; 2021). Additional heuristics and details are described in Supplement D.

### 3.1   Testing for memorization

We introduce four different tests to detect memorization. These tests use the fact that most tabular datasets have a canonical representation as a CSV file (which is nothing but a text document that might end up in the pre-training data). Intuitively, we say that a tabular dataset is memorized if the LLM can consistently generate it. In our context of tabular datasets, this is justified by the fact that these datasets contain random variables: It is impossible to consistently reproduce the realizations of random variables unless the values of the random variables have been seen before (i.e., during pre-training).[2]

Previous work on memorization in language models has often relied on the prefix-suffix pattern, where the model is provided with the initial part of a string as a prefix and asked to provide a completion (Carlini et al., 2019; 2021; 2022b). In this work, we use few-shot learning to condition chat models on the task of completing a given prefix, a prompt strategy

---

[2]This approach has been lightly formalized in Carlini et al. (2019) and Carlini et al. (2021), where the authors use the term 'canary'.

| | A. Knowledge and Learning | | | | B. Memorization | | | |
|---|---|---|---|---|---|---|---|---|
| | Feature Names | Feature Values | Feature Distribution | Conditional Distribution | Header Test | Row Compl. Test | Feature Compl. Test | First Token Test |
| Titanic | ✔/✔ | ✔/✔ | ✔/✔ | ✔/✔ | ✔/✔ | ✔/✔ | ✔/✔ | -/- |
| Adult | ✔/✔ | ✔/✔ | ✔/✔ | ✔/✔ | ✔/✔ | ✗/✗ | ✗/✗ | ✗/✗ |
| Diabetes | ✔/✔ | ✔/✔ | ✔/✔ | ✔/✔ | ✔/✔ | ?/✔ | ✔/✔ | ✗/✔ |
| Wine | ✔/✔ | ✔/✔ | ✔/✔ | ?/? | ✔/✔ | ?/✔ | ✔/✔ | -/- |
| Iris | ✔/✔ | ✔/✔ | ✔/✔ | ✔/✔ | ✔/✔ | ?/✔ | -/- | ?/✔ |
| Housing | ✔/✔ | ✔/✔ | ✔/✔ | ✔/✔ | ✔/✔ | ✗/✗ | ✗/✗ | -/- |
| Sp. Titanic | ✗/✗ | ✗/✗ | -/- | -/- | ✗/✗ | ✗/✗ | ✗/✗ | -/- |
| ACS Income | ✗/✗ | ✗/✗ | -/- | -/- | ✗/✗ | ✗/✗ | ✗/✗ | -/- |
| ICU | ✗/✗ | ✗/✗ | -/- | -/- | ✗/✗ | ✗/✗ | ✗/✗ | -/- |
| FICO | ✔/✔ | ✔/✔ | ?/? | ✗/✗ | ✗/✗ | ✗/✗ | ✗/✗ | ✗/✗ |
| ACS Travel | ✗/✗ | ✗/✗ | -/- | -/- | ✗/✗ | ✗/✗ | ✗/✗ | -/- |

Table 2: **GPT-3.5** and **GPT-4** have memorized many of the popular tabular datasets. The table depicts the results of different memorization tests with GPT-3.5 and GPT-4 (depicted in the table as */*). Test results are depicted in simplified form, that is, ✔= evidence of memorization, ✗= no evidence of memorization, ?= ambiguous result, and - = test cannot be conducted. The detailed, quantitative test results can be found in Supplement D.

that works well for GPT-3.5 and GPT-4, as well as for many other LLMs. The different tests can be performed with the tabmemcheck Python package. The tests are

1. The **Header Test:** We prompt the model with the initial rows of the CSV file and ask it to complete the next rows verbatim. This test measures the memorization of the initial rows of the dataset.

2. The **Row Completion Test:** We prompt the model with a number of contiguous rows from a random position of the CSV file and ask it to complete the next row verbatim. This

3. The **Feature Completion Test:** We prompt the model with all feature values of a random row in the dataset except for a single highly unique feature and ask it to complete the unique feature value verbatim. Examples of unique features are names and features with inherently high entropy, such as measurements with many decimal places.

4. The **First Token Test:** We prompt the model with a number of contiguous rows from a random position of the CSV file and ask it to complete the first token of the next row. If the rows of the CSV file are known to be random, we can perform a statistical test between the LLM completion accuracy and the accuracy of completion with the mode.

It is helpful to think about the proposed memorization tests in terms of power and significance, as we would about a statistical hypothesis test. All the tests are highly significant. This means they provide definitive evidence of memorization (though not necessarily of the entire dataset). Conversely, little can be said about the power of the tests. It seems possible that an LLM has memorized a tabular dataset, but we cannot extract it via prompting.

### 3.2 LLMs have memorized many of the popular tabular datasets

Table 2 shows the results of the memorization tests on 11 tabular datasets for GPT-3.5 and GPT-4. The Table depicts the results of the tests in simplified form, where ✔indicates that the model is able to generate the respective parts of the dataset. Quantitative test results and model generations can be found in Supplement D.

| | Titanic | Adult | Diabetes | Wine | Iris | Sp. Titanic | FICO |
|---|---|---|---|---|---|---|---|
| OLMo 7B | ?/ ✗ | ✗/ ✗ | ✗/ ✗ | ✗/ ✗ | ?/ ✓ | ✗/ ✗ | ✗/ ✗ |
| Gemma2 27B | ✓/ ✓ | ✓/ ✗ | ✓/ ✗ | ✓/ ✗ | ✓/ ✓ | ✗/ ✗ | ✗/ ✗ |
| Llama3 70B | ✓/ ✗ | ✓/ ✗ | ✓/ ✗ | ✓/ ✗ | ✓/ ✓ | ✗/ ✗ | ✗/ ✗ |
| Qwen1.5 72B | ✓/ ✗ | ✓/ ✗ | ✓/ ✗ | ✗/ ✗ | ✓/ ✓ | ✗/ ✗ | ✗/ ✗ |
| Llama3.1 405B | ✓/ ? | ✓/ ✗ | ✓/ ✗ | ✓/ ✗ | ✓/ ✓ | ?/ ✗ | ✗/ ✗ |

Table 3: **Open-source** and **open-weight** LLMs have memorized parts of the most popular tabular datasets. The table depicts the result of the **Header Test** and the **Row Completion Test** (depicted as \*/\*) in the simplified form used in Table 2. Detail in in Supplement D.

The *header test* indicates memorization of all the tabular datasets that were publicly available on the internet before 2021. This means that GPT-3.5 and GPT-4 have memorized the initial rows of these datasets. On `Titanic`, `Diabetes`, `Wine`, and `Iris`, the *row completion*, *feature completion*, and *first token* test also provide evidence of memorization. This means that GPT-3.5 and GPT-4 have not only memorized the initial rows of these datasets (which are frequently in Jupyter notebooks) but also random rows. Of course, this means that GPT-3.5 and GPT-4 have essentially memorized the entire datasets. We also observe that GPT-4 exhibits somewhat stronger evidence for memorization than GPT-3.5, which aligns with the literature that shows that larger models exhibit more memorization (Carlini et al., 2022a). Interestingly, there is no evidence of memorization of random rows on `Adult` and `Housing`.

On the novel datasets, there is no evidence of memorization for GPT-3.5 and GPT-4. This is to be expected for datasets released after the model was trained. In the case of the `FICO` dataset, it suggests that this dataset might have been protected from inclusion in the LLM training data by the need to register prior to access.[3]

Table 3 shows the results of the *header test* and the *row completion test* for five different open LLMs (Groeneveld et al., 2024; Riviere et al., 2024; Dubey et al., 2024; Qwen Team, 2024). Similar to the results observed in Table 2, there is evidence for the memorization of the most popular tabular datasets in almost all LLMs. However, with the exception of the Iris dataset, there is usually only evidence for the memorization of the initial rows and not of the entire dataset.

## 4 Few-Shot Learning with LLMs and Tabular Data

In the previous Section, we demonstrated that LLMs have memorized many popular tabular datasets. In this Section, we investigate the few-shot learning performance of GPT-3.5 and GPT-4 on the novel and memorized datasets. In addition, we replicate the entire analysis in this Section with Llama 3.1 70B and Gemma 2 27B. This can be found in Supplement F.

**Prompts.** We build on previous works and prompt chat models with tabular data in the format "Feature Name = Feature Value" (Borisov et al., 2023; Hegselmann et al., 2023).

---

**Few-Shot Learning with Tabular Data**

**System:** You are a classification assistant who is an expert in tabular data, data science, and cross-sectional population surveys. *[...]*

**User:** IF Age = 30, WorkClass = Private, fnlwgt = 196945, [...] THEN Income =
**Assistant:** >50k

*[few-shot examples]*

**User:** IF Age = 28, WorkClass = Self-emp-inc, fnlwgt = 79135, [...] THEN Income =
**Assistant:** [Model Response]

---

[3]Of course, it is possible that the LLM saw the dataset during training, but there was no memorization, or our memorization tests are not powerful enough.

`Original.` "You help to make predictions on the famous Iris flower data set."

|   | sepal_length | sepal_width | petal_length | petal_width | species |
|---|---|---|---|---|---|
| 0 | 5.1 | 3.5 | 1.4 | 0.2 | Iris-setosa |
| 1 | 4.9 | 3.0 | 1.4 | 0.2 | Iris-setosa |
| 2 | 4.7 | 3.2 | 1.3 | 0.2 | Iris-setosa |
| 3 | 4.6 | 3.1 | 1.5 | 0.2 | Iris-setosa |
| 4 | 5.0 | 3.6 | 1.4 | 0.2 | Iris-setosa |

`Task.` "You help the user to classify four different species of the Iris flowering plant."

|   | Length of Sepal (cm) | Width of Sepal (cm) | Length of Petal (cm) | Width of Petal (cm) | Kind of Flower |
|---|---|---|---|---|---|
| 0 | 5.21 | 3.43 | 1.30 | 0.19 | Setosa |
| 1 | 4.99 | 2.89 | 1.28 | 0.19 | Setosa |
| 2 | 4.81 | 3.30 | 1.17 | 0.19 | Setosa |
| 3 | 4.47 | 3.21 | 1.37 | 0.21 | Setosa |
| 4 | 4.92 | 3.50 | 1.30 | 0.20 | Setosa |

`Perturbed.` "You help to make predictions on the famous Iris flower data set."

|   | sepal_length | sepal_width | petal_length | petal_width | species |
|---|---|---|---|---|---|
| 0 | 5.2 | 3.4 | 1.3 | 0.2 | Iris-setosa |
| 1 | 5.0 | 2.9 | 1.3 | 0.2 | Iris-setosa |
| 2 | 4.8 | 3.3 | 1.2 | 0.2 | Iris-setosa |
| 3 | 4.5 | 3.2 | 1.4 | 0.2 | Iris-setosa |
| 4 | 4.9 | 3.5 | 1.3 | 0.2 | Iris-setosa |

`Statistical.` "You help the user with a classification problem, predicting the target variable from the inputs."

|   | X1 | X2 | X3 | X4 | Y |
|---|---|---|---|---|---|
| 0 | 2.65 | -2.76 | 4.69 | 4.33 | 0 |
| 1 | 3.43 | 1.30 | 4.71 | 4.37 | 0 |
| 2 | 4.22 | -1.73 | 4.91 | 4.40 | 0 |
| 3 | 5.54 | -1.12 | 4.46 | 4.30 | 0 |
| 4 | 3.70 | -3.34 | 4.59 | 4.24 | 0 |

Figure 1: Transformations of the `Iris` dataset. All datasets are presented to the LLM in four different formats: Original, perturbed, task, and statistical. See Section 4.1 for a description.

We ask the LLM to predict the value of the target variable, given the values of the other features in the data. We select 20 few-shot examples randomly and stratify the labels of the few-shot examples to match the label occurrence in the dataset. All experiments are conducted at temperature 0. A full example prompt is given in Supplement G.

## 4.1 Dataset Format: Original, Perturbed, Task and Statistical

We make use of an intriguing property of tabular data: It is possible to make changes to the format of the data without (significantly) affecting the underlying classification problem (compare Figure 1). We use this fact to explore (1) the consequences of memorization in Section 4.2 and (2) whether LLMs make use of their world knowledge in Section 4.3. In all few-shot learning experiments, we present the data in one of four standardized formats:

**Original** means that we present the data as it is contained in the CSV file of the dataset. In the **perturbed** version, we slightly change individual digits in the data that are not relevant to the underlying classification problem. We also deface any unique identifiers, such as observation IDs and names. In the system prompt of both original and perturbed, we tell the model about the origin of the data ("*You help to make predictions on the Titanic dataset from Kaggle*"). In the **task** version, we change the names of the features without changing their meaning ("BMI" becomes "Body mass index"), and similarly re-code categorical values ("0" becomes "False" and "United-States" becomes "USA"). We also round numerical values to two digits (unless this interferes with the meaning of a variable) and provide a generic system prompt ('You help to predict the type of a wine from its features."). In the **statistical** version, all numeric features are standardized to zero mean and constant variance. Feature names are replaced with X1, ..., Xn and strings encoded as categorical variables.

An important aspect of our datasets transforms is that they are standardized and comparable across datasets. Additional details on how we standardized the transformations are in Supplement E.

## 4.2 Memorization leads to inflated performance estimates

Table 4 depicts the few-shot learning performance of GPT-4 and GPT-3.5 on 10 different tabular datasets. On the memorized datasets depicted in Panel A of Table 4, the performance is quite impressive. In particular, GPT-4 outperforms logistic regression on 3 out of 5 datasets when prompted with the original data. However, the predictive performance of GPT-4 and GPT-3.5 drops as we move from the original data to the perturbed data. As depicted in Figures 11a and 11c, the accuracy of GPT-4 and GPT-3.5 on the memorized datasets consistently drops as we present the data in the original, perturbed, task and statistical

| Panel A | Titanic | | Adult | | Diabetes | | Wine | | Iris | |
|---|---|---|---|---|---|---|---|---|---|---|
| Original | $0.81_{01}$ | $0.96_{01}$ | $0.78_{01}$ | $0.81_{01}$ | $0.74_{02}$ | $0.74_{02}$ | $0.88_{02}$ | $0.96_{01}$ | $0.98_{01}$ | $0.99_{01}$ |
| Perturbed | $0.78_{01}$ | $0.82_{01}$ | $0.78_{01}$ | $0.81_{01}$ | $0.73_{02}$ | $0.73_{02}$ | $0.88_{02}$ | $0.95_{02}$ | $0.95_{02}$ | $0.95_{02}$ |
| Task | $0.77_{01}$ | $0.80_{01}$ | $0.75_{01}$ | $0.79_{01}$ | $0.70_{02}$ | $0.73_{02}$ | $0.87_{03}$ | $0.87_{03}$ | $0.95_{02}$ | $0.95_{02}$ |
| Statistical | $0.61_{02}$ | $0.65_{02}$ | $0.70_{01}$ | $0.63_{02}$ | $0.68_{02}$ | $0.62_{02}$ | $0.86_{03}$ | $0.90_{02}$ | $0.87_{03}$ | $0.92_{02}$ |
| LR / GBT | 0.79 / | 0.80 | 0.86 / | 0.87 | 0.78 / | 0.75 | 0.98 / | 0.96 | 0.97 / | 0.95 |

| Panel B | S. Titanic | | ACS Income | | ICU | | FICO | | ACS Travel | |
|---|---|---|---|---|---|---|---|---|---|---|
| Original | $0.58_{02}$ | $0.67_{01}$ | $0.78_{01}$ | $0.78_{01}$ | $0.69_{05}$ | $0.69_{05}$ | $0.58_{02}$ | $0.67_{01}$ | $0.54_{02}$ | $0.62_{02}$ |
| Perturbed | $0.57_{02}$ | $0.67_{01}$ | $0.77_{01}$ | $0.78_{01}$ | $0.69_{05}$ | $0.71_{05}$ | $0.58_{02}$ | $0.67_{02}$ | $0.54_{02}$ | $0.62_{02}$ |
| Task | $0.59_{02}$ | $0.65_{02}$ | $0.77_{01}$ | $0.77_{01}$ | $0.67_{05}$ | $0.71_{05}$ | $0.61_{02}$ | $0.68_{01}$ | $0.54_{02}$ | $0.65_{01}$ |
| Statistical | $0.63_{02}$ | $0.66_{01}$ | $0.59_{02}$ | $0.57_{02}$ | $0.56_{05}$ | $0.55_{05}$ | $0.60_{02}$ | $0.59_{02}$ | $0.52_{02}$ | $0.48_{02}$ |
| LR / GBT | 0.78 / | 0.78 | 0.80 / | 0.80 | 0.76 / | 0.66 | 0.70 / | 0.69 | 0.64 / | 0.67 |

Table 4: Few-shot learning performance of GPT-3.5 and GPT-4 across different tabular datasets. The table depicts the predictive accuracy and standard error of the LLMs on 10 different datasets. The table depicts the results with GPT-3.5 and GPT-4, separated by a space in the same column. Each dataset is presented to the LLM in four different formats: original, perturbed, task and statistical (compare Figure 1). **Panel A (top)** depicts results on datasets that the LLM has memorized. **Panel B (bottom)** depicts results on novel datasets where there is no evidence of memorization. The table also depicts the predictive accuracy of logistic regression (LR) and a gradient-boosted tree (GBT).

format. In particular, GPT-4 on the task data outperforms logistic regression only on 1 out of 5 datasets, and even there, the measured difference lies within the standard error.

Panel B of Table 4 depicts the few-shot learning performance of GPT-4 and GPT-3.5 on novel datasets. In contrast to the results in Panel A, GPT-4 in the original format does not outperform logistic regression on any dataset. Moreover, we also don't observe significant changes between the original, perturbed, and task formats. This is again depicted in Figures 11b and 11d. While it is always possible that the few-shot learning performance of an LLM varies with subtleties such as the length of a feature name or the number of significant digits of a number, the results in Panel B of Table 4 indicate that GPT-4 and GPT-3.5 are fairly robust to such changes.

The striking difference in performance between the different dataset formats on the memorized datasets, which is completely absent on the novel datasets, strongly suggests that memorization leads to invalid performance estimates of few-shot learning. Even though the datasets in Panel A and Panel B of Table 4 vary along other dimensions than being memorized or novel, it seems highly unlikely that these give rise to the observed performance differences with logistic regression or the observed performance drops between the different dataset versions.

We note that the GPT-4 results in Table 4 were obtained with `gpt-4-0125-preview`, which has seen data up to December 2023. On the memorization tests, `gpt-4-0125-preview` obtains the same results as the GPT-4 models with a training data cutoff in 2021.[4]

### 4.3 Performance depends on feature names and variable format

While the results in Panel A of Table 4 indicate that few-shot learning on datasets seen during training can suffer from overfitting, Panel B demonstrates that GPT-3.5 and GPT-4 perform reasonably well on novel data.

---

[4]Producing the results in Table 4 with `gpt-4-32k-0613` would have been prohibitively expensive. We conducted limited experiments with `gpt-4-32k-0613` on the original data format where the few-shot learning performance matched `gpt-4-0125-preview`.

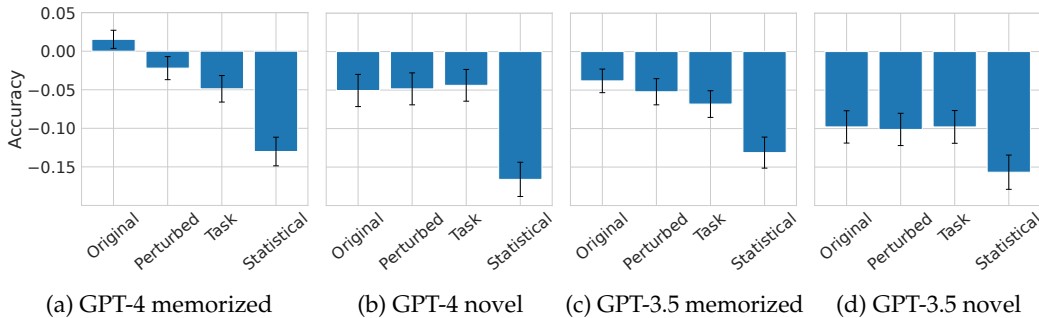

(a) GPT-4 memorized     (b) GPT-4 novel     (c) GPT-3.5 memorized     (d) GPT-3.5 novel

Figure 2: Few-shot learning performance of GPT-4 and GPT-3.5 across memorized and novel datasets. The Figure depicts the absolute performance difference between the LLM and logistic regression, that is, a value of 0 means that the LLM has the same accuracy as logistic regression. The Figure depicts the average accuracy across the different datasets. Parentheses indicate standard errors. This Figure summarizes the results in Table 4.

Moreover, on all datasets except Spaceship Titanic, the performance is consistent across the original, perturbed, and task formats but significantly drops for the statistical format. This indicates that the LLMs rely on the natural scale of the feature values and the variable names, which are made unrecognizable in the statistical format. We suggest that this effect is not observed on Spaceship Titanic because this dataset is synthetically generated and about a fictional event with fictional variable names.

This effect is also depicted in Figure 2, where there is a significant drop in accuracy from the task to statistical setting in all subfigures.

### 4.4 Language models act as statistical predictors, but lag behind traditional algorithms

In the previous section, we have seen a significant performance drop between task and statistical, indicating that the few-shot learning performance of LLMs significantly relies on the LLM's world knowledge (Yu et al., 2023). In this section, we study the ability of language models to act as in-context statistical learners. This relates to works using small, specifically trained language models to show that in-context learning can learn simple function classes, akin to supervised learning (Garg et al., 2022; Von Oswald et al., 2023). We investigate the question in the context of GPT-3.5 and GPT-4, similarly to Bhattamishra et al. (2024).

**Binary Statistical Classification.** We study a simple statistical learning problem with numerical features $x_i \sim \mathcal{N}(0, I) \in \mathbb{R}^d$, an unknown coefficient vector $z \sim \mathcal{N}(0, I) \in \mathbb{R}^d$, and binary labels $y_i = (x_i^T z >= 0)$. The data is presented to the model in the statistical format (see Figure 1).

**Few-shot learning performance degrades in the dimension of the problem.** Figure 3a depicts the few-shot learning performance on our binary statistical classification problem for a fixed number of 20 few-shot examples as the dimension of the problem increases. For $d = 2$, the performance of GPT-3.5 and GPT-4 is roughly on par with the 1-Nearest Neighbor classifier and logistic regression. As the dimension of the problem increases, the few-shot learning performance of the LLMs deteriorates, and more quickly so than for the traditional statistical learning algorithms.

**Few-shot learning performance of GPT-4 scales in the number of few-shot examples.** Figure 3b depicts the few-shot learning performance on our binary statistical classification problem for the fixed dimension of 8 and an increasing number of few-shot examples. Whereas GPT-3.5 scales only very weakly in the number of few-shot examples, the performance of GPT-4 increases monotonically. For 200 few-shot examples, GPT-4 is roughly on par with the 1-Nearest Neighbor classifier, but remains less efficient than logistic regression.

**Fine-tuning scales in the number of few-shot examples.** It is interesting to compare the scaling in the number of few-shot examples for in-context learning versus fine-tuning.

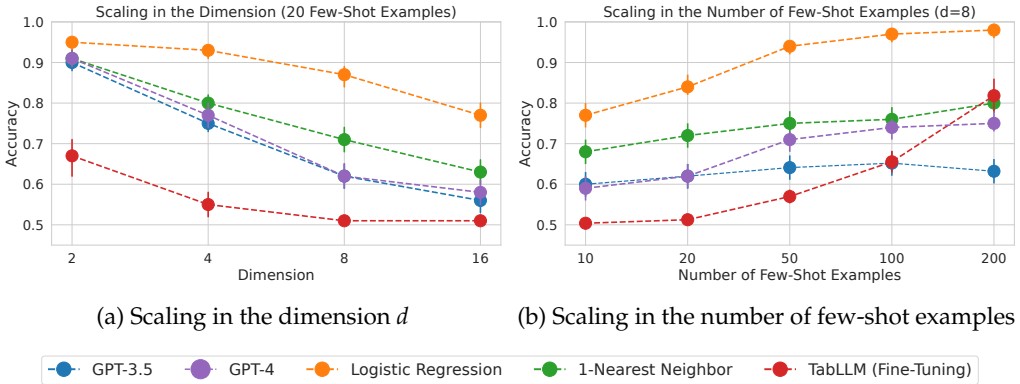

(a) Scaling in the dimension $d$
(b) Scaling in the number of few-shot examples

Figure 3: Few-shot learning performance of GPT-3.5, GPT-4, TabLLM (fine-tuning a language model with 11B parameters), Logistic Regression and a 1-Nearest Neighbor classifier across binary classification problems with a linear decision boundary. Figure (a) depicts the scaling of the few-shot learning performance in the dimension of the problem (that is, the number of features). We use 20 few-shot examples across all dimensions. Figure (b) depicts the scaling of the few-shot learning performance in the number of few-shot examples (respectively, the size of the training set). We use a fixed dimension of 8. Mean and 95% confidence intervals.

Figure 3b depicts the performance of the fine-tuning technique TabLLM by Hegselmann et al. (2023) on our binary statistical classification problem. We see that the performance of the fine-tuning approach scales in the number of samples.

## 5 Drawing Random Samples from Datasets seen during Training

In the previous Section, we explored the implications of memorization on few-shot learning with tabular data. However, predicting the target label is only one of many tasks that one might want to perform with LLMs and tabular data (Hollmann et al., 2024; Bordt et al., 2024; Sui et al., 2024). In this section, we demonstrate another implication of seeing a dataset during training: We show that LLMs can draw random samples from these datasets – without any fine-tuning (Borisov et al., 2023).

Figure 4 depicts the longitude and latitude of random samples on the California Housing dataset. To draw these samples, we conditioned GPT-3.5 with samples from *other* datasets,

Figure 4: GPT-3.5 can draw random saFiguremple from the California Housing dataset. We only provide the model with the name of the dataset and the feature names. The diversity of the generated samples depends on the temperature parameter. For small temperatures, the samples are concentrated around the mode of data. As temperature increases, the samples become more diverse and similar to the data distribution. At large temperatures, some samples lie outside the support of the data distribution. The reader might want to compare with Figure 1 in Borisov et al. (2023).

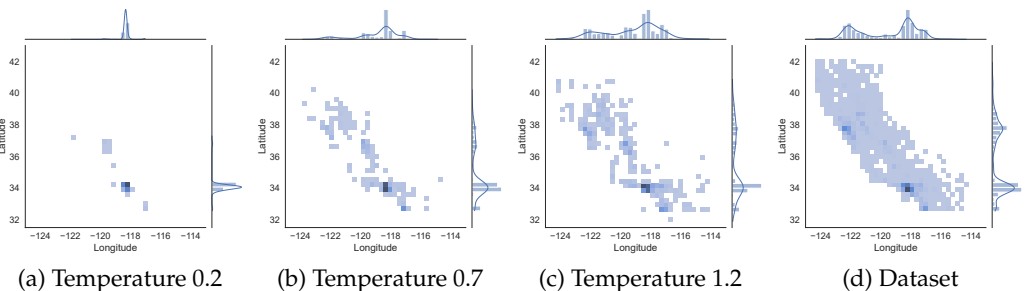

(a) Temperature 0.2
(b) Temperature 0.7
(c) Temperature 1.2
(d) Dataset

revealing no information about the feature values on the California Housing dataset. Interestingly, the samples drawn by the LLM follow the overall summary statistics in the data. For example, the feature correlations of the samples match the feature correlation in the original dataset (Supplement Figure 16). At the same time, the samples are not copied from the training data (Supplement Table 11). In this sense, it can be said that GPT-3.5 generates novel random samples from the dataset.

## 6 Related Work

Recent works have demonstrated the capabilities of LLMs on tasks with **tabular data** (Dinh et al., 2022; Narayan et al., 2022; Vos et al., 2022; Wang et al., 2023; McMaster et al., 2023). In particular, Borisov et al. (2023) and Hegselmann et al. (2023) have shown that LLMs can be fine-tuned to generate and classify tabular data. The phenomenon of **in-context learning** has attracted much research interest (Brown et al., 2020; Wei et al., 2023; Li et al., 2023). It has been shown that the forward pass of a transformer can mimic gradient descent (Von Oswald et al., 2023) and that LLMs can learn simple function classes in-context (Garg et al., 2022; Bhattamishra et al., 2024).

Data or task **contamination** is studied in many LLM reports and research papers, often adopting an n-gram-based definition (OpenAI, 2023; Nori et al., 2023; Touvron et al., 2023). Jiang et al. (2024) investigate data contamination in GPT-2 by adding benchmark datasets to the pre-training data. Yang et al. (2023) study the effects of rephrasing widely used benchmarks. Li & Flanigan (2024) analyze how the few-shot learning performance of LLMs varies between datasets released before and after the training data was collected.

**Membership inference attacks** are a set of methods to detect if a model has seen a text during training (Mahloujifar et al., 2021; Choquette-Choo et al., 2021; Carlini et al., 2022a; Mireshghallah et al., 2022; Shi et al., 2024; Mattern et al., 2023). Recently, Duan et al. (2024) question the value of membership inference attacks for large language models. Shi et al. (2024) introduce the Min-K% Prob method to detect pre-training data from LLMs.

Research on **memorization** has shown that LLMs can be prompted to repeat chunks of their training data verbatim (Carlini et al., 2019; 2021; Chang et al., 2023; Nasr et al., 2023). Memorization is linked to data-duplication in the pre-training data (Carlini et al., 2022b; Biderman et al., 2023). Memorization does not necessarily lead to overfitting (Magar & Schwartz, 2022).

LLMs tend to perform better on **tasks more likely to occur in the training data** (Wu et al., 2023; McCoy et al., 2023; Dziri et al., 2023b). Our work complements this emerging literature by (1) rigorously demonstrating that GPT-3.5 and GPT-4 have seen certain tasks frequently during training (whereas previous works have relied on heuristics) and (2) showing that LLMs overfit on uncommon tasks that were seen frequently during training.

## 7 Discussion

In this work, we have first shown that LLMs have memorized many of the popular tabular datasets. We have then used this fact to gauge the amount of overfitting that occurs during in-context learning with such datasets. We find strong evidence of overfitting in in-context learning due to memorization. We have also studied other factors that determine the few-shot learning performance of GPT-3.5 and GPT-4 on prediction tasks with tabular data, finding that these models rely on their world knowledge, but also have the ability to act as statistical predictors.

Our study has several limitations, such as that we usually don't know if and in what way the LLMs saw the different tabular datasets during training. Also, it would have been more ideal had we been able to insert datasets into the training corpus at random.

Finally, we want to highlight that had we performed our evaluations without carefully controlling for memorization, we might have erroneously concluded that few-shot learning with GPT-4 frequently outperforms traditional statistical learning algorithms.

## 8 Ethics Statement

All the datasets we use in this study are explicitly available for scientific research. Because our study concerns the scientific question of invalid performance evaluations of LLMs, we don't believe it raises significant ethical concerns. While we consider memorization purely from the perspective of training data contamination, it has broader implications, including copyright and privacy.

## 9 Reproducibility Statment

We conducted initial experiments with different versions of GPT-3.5 and GPT-4 and found that the results are fairly robust towards the precise model version. This holds true for both the results of the memorization tests in Table 2 and for the few-shot learning results in Table 4. An exception is the model `gpt-3.5-turbo-1106` that performs worse on the few-shot learning tasks than other versions of GPT-3.5. The models that we used to run the final experiments are detailed in Supplement B. The cost of replicating all the results in this paper with the Open AI API is approximately 1000 USD. By far, the most expensive experiments are the few-shot learning experiments with GPT-4 (they require approximately 1000 queries per data point, sometimes with relatively long context). In contrast, the memorization tests require relatively few queries.

Code to replicate the results in this paper is available at `https://github.com/interpretml/LLM-Tabular-Memorization-Checker`. This includes, in particular, the different memorization tests and the dataset transformations.

## Acknowledgments

Sebastian is supported by the Tübingen AI Center and the German Research Foundation through the Cluster of Excellence "Machine Learning - New Perspectives for Science" (EXC 2064/1 number 390727645). This project was started while Sebastian was an intern at Microsoft Research. The authors would like to thank Varun Chandrasekaran for helpful discussion.

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

# A  Datasets

**Iris.** The Iris flower dataset is a small dataset (150 observations) that is freely available on the Internet, among others in the UCI repository at `https://archive.ics.uci.edu/dataset/53/iris`.

**Wine.** The Wine dataset is another small dataset from the UCI repository (178 observations). It is available at `https://archive.ics.uci.edu/ml/machine-learning-databases/wine/wine.data`.

**Kaggle Titanic.** The Kaggle Titanic dataset is a popular and freely available machine learning dataset. It is available at `https://www.kaggle.com/competitions/titanic`.

**OpenML Diabetes.** The OpenML Diabetes dataset (Smith et al., 1988) is a popular dataset that is freely available `https://www.openml.org/search?type=data&sort=runs&id=37&status=active` as part of OpenML (Vanschoren et al., 2014).

**Adult Income.** Historically, the Adult Income dataset is one of the most popular machine learning datasets `http://www.cs.toronto.edu/~delve/data/adult/adultDetail.html`. Recently, researchers have argued that this dataset should no longer be used Ding et al. (2021). The csv file of the dataset can still be found at Kaggle `https://www.kaggle.com/datasets/wenruliu/adult-income-dataset` and at many other places.

**California Housing.** The California Housing dataset (Kelley Pace & Barry, 1997) is a freely available and very popular machine learning dataset with 20640 observations.

**FICO.** The FICO HELIOCv1 dataset was part of the FICO Explainable Machine Learning Challenge `https://community.fico.com/s/explainable-machine-learning-challenge`. This dataset can be obtained only after signing a data usage agreement and is then available via Google Drive. This is an example of a dataset that is freely available but where the CSV file has not been publicly released on the internet, at least not by the creators of the dataset.

**Spaceship Titanic.** The Spaceship Titanic dataset was released on Kaggle in 2022 `https://www.kaggle.com/c/spaceship-titanic`. It is a synthetically generated dataset.

**ACSIncome** and **ACSTravel**. We created these two datasets using the `folktables` package and Census Data from the 2022 American Community Survey (ACS). The construction of the datasets is described in (Ding et al., 2021).

**ICU.** We created the UCI dataset as an example of a novel tabular dataset with few observations and features, similar to some of the traditional machine learning datasets. We created the dataset from the data at `https://dataverse.harvard.edu/dataset.xhtml?persistentId=doi:10.7910/DVN/PX9K2R`. The dataset has 10 features (patient characteristics) and a binary target. It has 102 observations.

# B  Models

Here provide the details of the models that were used to generate the results in the paper.

**Table 1:** `gpt-3.5-turbo-0613` and `GPT-4-32k-0315` deployed on Azure.

**Table 2:** `gpt-3.5-turbo-16k-0613` and `gpt-4-0613` deployed on Azure, with the exception of `gpt-3.5-turbo-0125` and `gpt-4-0125-preview` with the OpenAI API for the ACS Income, ACS Travel and ICU datasets. We also ran the memorization tests with `gpt-3.5-turbo-16k-0613` and `gpt-4-0125-preview` with the OpenAI API and obtained similar results. The main difference is that `gpt-4-0125-preview` knows the Spaceship Titanic dataset's feature names and feature values (but there is no evidence of memorization from our tests).

**Table 3:** `gpt-3.5-turbo-0125` and `gpt-4-0125-preview` with the OpenAI API.

**Figure 3:** `gpt-3.5-turbo-0125` with the OpenAI API and `gpt-4-0125-preview` deployed on Azure.

**Figure 4:** `gpt-3.5-turbo-0613` deployed on Azure.

**Logistic Regression.** We train logistic regression using scikit-learn (Pedregosa et al., 2011). We cross-validate the $L_2$ regularization constant.

**Gradient Boosted Tree.** We train gradient boosted trees using xgboost (Chen & Guestrin, 2016). We cross-validate the max_depth of the trees and the $L_1$ and $L_2$ regularization constants, similar to (Hegselmann et al., 2023).

## C  Time Series Experiments

For each year, we compute the robust mean relative error as

$$\frac{1}{T-20} \sum_{t=20}^{T} \min\{\frac{|\hat{y}_t - y_t|}{|y_t|}, 0.02\}. \tag{1}$$

We use a simple system prompt.

**System:** "You are an expert financial market analyst and stock trader and your task is to predict the development of the MSCI World stock index in 2022. You are given the price level of the stock index in the previous 20 days and predict the price of the stock index on the current day."

## D  Memorization Tests

This Section provides additional details on the memorization tests and their quantitative results. We also detail a number of measures that assess what the LLM knows about the dataset (for example, the feature names).

**With regard to the interpretation of the memorization tests**, it is important to note that the quantitative test results (e.g., how many rows the LLM managed to complete in the row completion test) are not sufficient to judge memorization. Instead, one has to consider the completion rate of the LLM and the amount of entropy in the dataset. For example, the OpenML Diabetes dataset contains an individual's glucose level, blood pressure, and BMI, as well as other measurements that are at least in part random and highly unique. If an LLM can consistently generate even a few rows of this dataset, it is strong evidence of memorization. To give another example, the Iris dataset contains many rows that are near-duplicates. This means that an LLM might also achieve a non-zero row completion rate by chance or prediction. Because of this complication, we manually judge the results in Table 2 and Table 3 in the main paper, considering the quantitative results of the different tests and the structure of the dataset.

As a general observation, one might expect the results of the tests to be somewhat ambiguous. It turns out, however, that GPT-3.5 and GPT-4 (as well as other LLMs) have the remarkable property of regurgitating quite long and complex strings from their training data. For example, GPT-4 is able to generate the entire Iris dataset, consisting of 151 rows, given only the first couple of rows as prompt. Thus, while there are few cases of ambiguity (marked as question marks in the respective tables), the evidence of memorization is usually fairly clear (and conversely, there are cases where the model gets the dataset's content completely wrong).

All memorization tests are conducted at temperature 0.

### D.1  Qualitative Examples

Figures 5-10 on the following pages illustrate the different memorization tests. They depict the Leveshtein string distance between the model response and the actual content in the dataset. For the different tests, we depict two cases: one with evidence of memorization and one without.

```
header_prompt, header_completion, response = tabmemcheck.header_test('datasets/tabular/adult-train.csv', "gpt-4-0125-preview")
✓ 1m 45.6s
```

```
Header Test: Age,WorkClass,fnlwgt,Education,EducationNum,MaritalStatus,Occupation,Relationship,Race,Gender,CapitalGain,CapitalLoss,H
39, State-gov,77516, Bachelors,13, Never-married, Adm-clerical, Not-in-family, White, Male,2174,0,40, United-States, <=50K
50, Self-emp-not-inc,83311, Bachelors,13, Married-civ-spouse, Exec-managerial, Husband, White, Male,0,0,13, United-States, <=50K
38, Private,215646, HS-grad,9, Divorced, Handlers-cleaners, Not-in-family, White, Male,0,0,40, United-States, <=50K
53, Private,234721, 11th,7, Married-civ-spouse, Handlers-cleaners, Husband, Black, Male,0,0,40, United-States, <=50K
28, Private,338409, Bachelors,13, Married-civ-spouse, Prof-specialty, Wife, Black, Female,0,0,40, Cuba, <=50K
37, Private,284582, Masters,14, Married-civ-spouse, Exec-managerial, Wife, White, Female,0,0,40, United-States, <=50K
49, Private,160187, 9th,5, Married-spouse-absent, Other-service, Not-in-family, Black, Female,0,0,16, Jamaica, <=50K
52, Self-emp-not-inc,209642, HS-grad,9, Married-civ-spouse, Exec-managerial, Husband, White, Male,0,0,45, United-States, >50K
31, Private,45781, Masters,14, Never-married, Prof-specialty, Not-in-family, White, Female,14084,0,50, United-States, >50K
42, Private,159449, Bachelors,13, Married-civ-spouse, Exec-managerial, Husband, White, Male,5178,0,40, United-States, >50K
Header Test Legend:  Prompt Correct Incorrect Missing
```

Figure 5: **Header Test** on **Adult Income**. The LLM is prompted with the first couple of rows of the dataset (black text) and responds with the next 7 rows of the dataset, exactly as they appear in the CSV file of the dataset (green text). The text color illustrates the Levenshtein string distance between the text in the CSV file and the model response. An entirely green row means that the model responded with the next row exactly as it occurs in the CSV file. Red color indicates that the model made a mistake, and violet color indicates that the model missed a digit. In this particular example, the model response is all green because it is equal to the content of the CSV file. Best viewed in digital format.

```
rows, responses = tabmemcheck.row_completion_test('iris.csv', 'gpt-4-0125-preview', num_queries=25)
```

```
Info: 1.99% of the rows in this dataset are duplicates.
5,3.5,1.3,0.3,Iris-setosa
5.9,3.2,4.8,1.8,Iris-versicolor
6.9,3.2,5.7,2.3,Iris-virginica
5.7,3.8,1.7,0.3,Iris-setosa
6.7,3.1,5.6,2.4,Iris-virginica
5.5,2.5,4.9,1.3,Iris-versicolor
6.3,2.8,5.1,1.5,Iris-virginica
6.4,3.2,4.5,1.5,Iris-versicolor
7.3,2.9,6.3,1.8,Iris-virginica
6,2.2,5,1.5,Iris-virginica
6.1,2.6,5.6,1.4,Iris-virginica
4.8,3.4,1.9,0.2,Iris-setosa
6.3,2.7,4.9,1.8,Iris-virginica
6.8,3.2,5.9,2.3,Iris-virginica
6.3,3.3,4.7,1.6,Iris-versicolor
5.9,3,4.2,1.5,Iris-versicolor
4.4,3.2,1.3,0.2,Iris-setosa
6.3,2.9,5.6,1.8,Iris-virginica
5.2,4.1,1.5,0.1,Iris-setosa
6.7,3,5,1.7,Iris-versicolor
5.7,4.4,1.5,0.4,Iris-setosa
5,3.5,1.6,0.6,Iris-setosa
7.1,3,5.9,2.1,Iris-virginica
6,2.7,5.1,6,1.6,Iris-versicolor
5.5,2.6,4.4,1.2,Iris-versicolor
Row Completion Test: 22/25 exact matches.
Legend:  Prompt Correct Incorrect Missing
```

Figure 6: **Row Completion Test** on **Iris**. The LLM is prompted with the previous rows of the dataset and responds with the next row. The figure depicts the Levenshtein string distance between the model responses and the actual next rows in the CSV file. An entirely green row means that the model responded with the next row exactly as it occurs in the CSV file. Red color indicates that the model made a mistake, and violet color indicates that the model missed a digit. Best viewed in digital format.

```
header_prompt, header_completion, response = tabmemcheck.header_test('datasets/tabular/acs-income-2022.csv', "gpt-4-0125-preview")
✓ 1m 27.1s
```

```
Header Test: Age,Class of worker,Educational attainment,Marital status,Occupation,Place of birth,Usual hours worked per week past 12 months,Sex,Recoded race,Income
26,"Employee of a private for-profit company or business, or of an individual, for wages, salary, or commissions","1 or more years of college credit, no degree",Never married or under
38,Federal government employee,Regular high school diploma,Divorced,Photographers,Arizona/AZ,40,Female,White alone,"Less than $50,000 per year."
23,Federal government employee,Regular high school diploma,Never married or under 15 years old,Military Enlisted Tactical Operations And Air/Weapons Specialists And Crew Members,Monta
20,"Employee of a private for-profit company or business, or of an individual, for wages, salary, or commissions","1 or more years of college credit, no degree",Never married or under
20,Federal government employee,Regular high school diploma,Never married or under 15 years old,"Military, Rank Not Specified",Tennessee/TN,50,Female,Two or More Races,"Less than $50,0
20,Federal government employee,Regular high school diploma,Married,Military Enlisted Tactical Operations And Air/Weapons Specialists And Crew Members,Illinois/IL,40,Male,White alone,"
23,"Employee of a private for-profit company or business, or of an individual, for wages, salary, or commissions","Bachelor's degree",Never married or under 15 years old,"LaSorers Afl
35,FeState government employee,ReguMar high ster's diplomegrever,Married,Secondar 15 y School Teachenters,New York/NY,40,Female,Black or African American alone,"$50,000 or more per ye
28,Self-employed of a privatn fown not incorporated business, wor okers inwithoualt employees,Associate degree,Divor ced,Real Estate Brokers,Florida/FL,60,Female,White alone,"Less tha
45,Local government employee,Doctorate or professiondarl degree,Married,Physicians,Ohio/OH,50,Male,White alone,"Le$50,000 or more per year."
32,"Employee of a private for-profit company or business, or of an individual, for wages, salary, or commissions","Some 11,hivorced,gh scellahool, Pnodu dionplorkma",Married,Construl
29,Federal government employee,ReguBar high school or's degrever,Married o,Civil Engineers,Virginia/VA,40,Male,Asian alone,"$50,000 or more per year."
27,Self-employed in own inces,orporathed $5business, ywor."
30,kederas gwith paid employees,Bachelor's degree,Never married, ors under 15 years old,Graphic Designers,Coliforado/CO,50,Female,SomWhither Race alone,"Mo$50,000 or more per year."
34,Unpaid family worker,Regular high school diploma,Married,"Not in labor force",Utah/UT,0,Female,White alone,"Less than $50,000 per year."
Header Test Legend:  Prompt Correct Incorrect Missing
```

Figure 7: **Header Test** on **ACS Income**. The LLM is prompted with the first couple of rows of the dataset (black text) and responds with 8 more rows. The text color illustrates the Levenshtein string distance between the text in the CSV file and the model response. An entirely green row means that the model responded with the next row exactly as it occurs in the CSV file. Red color indicates that the model made a mistake, and violet color indicates that the model missed a digit. Best viewed in digital format.

```
tabmemcheck.row_completion_test('datasets/tabular/heloc_dataset_v1.csv', "gpt-4-0125-preview")
✓ 1m 50.1s

Info: 5.61% of the rows in this dataset are duplicates.
Bad,70,210,5,72,18,0,0,100,-7,7,8,19,12,45,0,2,2,85,95,3,3,1,185
Good,72,210,5,98,20,0,0,100,-7,7,8,21,1,37,-7,0,0,18,55,4,2,0,64
Good,75,210,7,105,37,0,0,100,-7,7,8,40,2,30,-7,0,0,53,55,1,2,0,75
GBad,65,342,10,258,70,14,0,100,85,5,4,6,18,2,40,0,2,12,39,85,4,2,1,75
Bad,67,210,4,60,14,1,1,88,5,4,6,20,2,40,0,2,2,72,85,3,2,1,175
Bad,70,190,4,88,18,1,1,92,12,4,4,19,1,27,3,0,0,15,67,2,3,1,39
GBad,70,210,3,78,22,1,1,89,34,6,6,25,12,33,0,1,1,67,89,3,3,2,67
Bad,59,131,7,85,26,0,0,88,2,4,4,28,1,33,0,0,0,18,175,5,2,2,67
Good,88,210,5,88,25,0,0,100,-7,7,8,26,2,20,-7,0,0,5,101,3,2,0,67
Good,75,259,98,101,22,0,0,197,12,6,6,21,2,018,0,1,1,7,83,4,-2,0,155
Bad,59,218,17,70,28,0,0,95,34,4,6,28,1,40,0,0,0,72,66,15,3,2,80
Good,82,215,8,188,22,0,0,100,-7,7,8,22,3,31,0,1,1,12,79,3,2,0,55
Good,78,215,7,98,22,0,0,100,-7,7,8,22,2,37,0,0,0,63,-8,12,2,0,59
Bad,59,190,7,68,19,0,0,88,15,4,4,21,2,33,0,2,2,76,83,4,3,2,89
Good,78,210,3,78,32,0,0,100,-7,7,8,35,3,24,0,3,3,58,85,4,3,0,65
Good,67,2510,4,89,33,0,0,100,-7,7,8,35,2,30,0,2,2,15,85,3,3,0,45
GBad,70,211,4,87,23,0,0,100,-7,7,8,23,2,37,0,1,1,34,77,3,3,1,74
GBad,67,214,7,56,23,0,0,100,-7,7,8,23,1,45,0,0,0,38,76,3,2,0,59
Good,75,210,2,65,22,0,0,100,-7,7,8,23,2,40,0,1,1,35,88,3,2,0,55
Good,70,100,7,120,14,0,0,100,-7,7,8,14,2,20,-7,0,0,30,-8,3,-8,0,75
Good,-9,-9,-9,-9,-9,-9,-9,-9,-9,-9,-9,-9,-9,-9,-9,-9,-9,-9,-9,-9,-9,-9,-9
Good,67,180,3,78,33,0,0,100,-7,7,8,35,2,29,4,1,1,69,92,3,2,0,54
Good,68,210,5,68,27,0,0,100,-7,7,8,29,4,39,20,3,3,45,76,3,3,1,55
Bad,75,142,5,57,18,2,2,187,12,6,3,20,2,37,0,1,1,78,85,5,4,1,74
Good,72,198,23,88,32,0,0,100,-7,7,8,34,2,29,4,1,1,9,92,2,2,0,39
Row Completion Test: 1/25 exact matches.
Legend:  Prompt Correct Incorrect Missing
```

Figure 8: **Row Completion Test** on **FICO**. The LLM is prompted with the previous rows of the dataset and responds with the next row. The text color illustrates the Levenshtein string distance between the text in the CSV file and the model response. An entirely green row means that the model responded with the next row exactly as it occurs in the CSV file. Red color indicates that the model made a mistake, and violet color indicates that the model missed a digit. Best viewed in digital format.

```
feature_values, responses = tabmemcheck.feature_completion_test('datasets/tabular/openml-diabetes.csv', "gpt-4-0125-preview", num_queries=25)
✓  1m 14.7s
```
Info: Using feature DiabetesPedigreeFunction with 67.32% unique values.
0.696
0.187
0.38
0.121
0.341
0.845
0.855
0.304
0.331
0.532
0.315
0.370
0.466
0.499
0.233
0.557
0.717
0.828
0.259
0.138
0.412
0.804
0.183
0.378
0.300
Legend:  Prompt Correct Incorrect Missing

Figure 9: **Feature Completion Test** on **OpenML Diabetes**. The LLM is prompted with the feature values of a row, except for a single missing feature. The model responds with the value of the missing feature. The figure depicts the Levenshtein string distance between the model responses and the actual feature values in the CSV file. An entirely green text means that the model responded with the feature value exactly as it occurs in the CSV file. Red color indicates that the model made a mistake, and violet color indicates that the model missed a digit. Best viewed in digital format.

```
feature_values, responses = tabmemcheck.feature_completion_test('datasets/tabular/adult-train.csv', "gpt-4-0125-preview", num_queries=25)
✓  50.6s
```
Info: Using feature fnlwgt with 66.48% unique values.
196689
168294
132222
122272
234721
234721
164526
209642
202683
5234721
193884
192762
176261
188300
189346
168294
41046907
151910
286558
141297
168294
209642
292175
1208103
102864
Legend:  Prompt Correct Incorrect Missing

Figure 10: **Feature Completion Test** on **Adult Income**. The LLM is prompted with the feature values of a row, except for a single missing feature. The model responds with the value of the missing feature. The figure depicts the Levenshtein string distance between the model responses and the actual feature values in the CSV file. An entirely green text means that the model responded with the feature value exactly as it occurs in the CSV file. Red color indicates that the model made a mistake, and violet color indicates that the model missed a digit. Best viewed in digital format.

|  | **Row Completion Test** | | **Feature Completion Test** | | |
|  | GPT-3.5 | GPT-4 | GPT-3.5 | GPT-4 | Feature Name |
|---|---|---|---|---|---|
| Iris | 35 / 136 | 125 / 136 | - | - | - |
| Wine | 16 / 164 | 84 / 164 | 77 / 178 | 131 / 178 | malic_acid |
| Kaggle Titanic | 194 / 250 | 222 / 250 | 238 / 250 | 236 / 250 | Name |
| OpenML Diabetes | 18 / 250 | 79 / 250 | 237 / 250 | 243 / 250 | DiabetesPedigreeFunction |
| Adult Income | 0 / 250 | 0 / 250 | 0 / 250 | 0 / 250 | fnlwgt |
| California Housing | 0 / 250 | 0 / 250 | 0 / 250 | 1 / 250 | median_income |
| FICO | 1 / 250 | 2 / 250 | 2 / 250 | 14 / 250 | MSinceOldestTradeOpen |
| Spaceship Titanic | 0 / 250 | 0 / 250 | 0 / 250 | 2 / 250 | Name |
| ICU | 0 / 25 | 0 / 25 | 0 / 25 | 0 / 25 | Glucose |
| ACS Income | 0 / 25 | 0 / 25 | 2 / 25 | 2 / 25 | Occupation |
| ACS Travel | 0 / 25 | 0 / 25 | 2 / 25 | 0 / 25 | Occupation |

Table 5: Quantitative test results of the row completion and feature completion tests from Table 2 in the main paper. The table depicts the number of rows and features that were correctly completed.

|  | **First Token Test** | | |
|  | GPT-3.5 | GPT-4 | Baseline Accuracy (Best of Mode, LR and GBT) |
|---|---|---|---|
| Iris | 88 / 136 (0.65) | 131 / 136 (0.96) | 0.50 |
| Wine | - | - | 0.95 |
| Kaggle Titanic | - | - | - |
| OpenML Diabetes | 42 / 250 (0.17) | 95 / 250 (0.38) | 0.25 |
| Adult Income | 59 / 250 (0.24) | 68 / 250 (0.27) | 0.26 |
| California Housing | - | - | 0.95 |
| Spaceship Titanic | - | - | - |
| ICU | - | - | - |
| ACS Income | - | - | - |
| FICO | 119 / 250 (0.48) | 78 / 250 (0.31) | 0.47 |
| ACS Travel | - | - | - |

Table 6: Quantitative test results of first token test from Table 2 in the main paper. The table depicts the number of first tokens that were correctly completed and the overall accuracy of the completions (in brackets). The last row depicts the completion accuracy that can be reached using traditional statistical predictors (this improves upon the mode if the rows are not random).

## D.2 Quantitative Test Results

Table 5 and Table 6 depict the quantitative results of the row completion, feature completion, and first token test.

For the row completion test depicted in Table 5, we count the number of times that a row is correctly completed. In comparison with Figure 6 and Figure 8, which depict model responses on the row completion task, this means that we count the number of entirely green rows (i.e. zero Levenshtein distance between the row in the CSV file and the model response). From the numbers depicted in Table 5, we see that the number of row completions is either zero / in very low digits or fairly large, with over 90% correctly completed rows on some datasets. As mentioned before, the number of correct row completions has to be judged in comparison with the amount of entropy in the data. A concrete example of this

is depicted in Figure 8, where the correctly completed row on FICO is one of the many duplicate rows in the dataset that consists repeatedly of the number -9. Of course, this does not provide evidence of memorization.

For the feature completion test in Table 5, we count the number of times that a feature value is correctly completed. We chose the feature to be completed as a highly unique feature (the Iris dataset does not contain such a feature, so we don't conduct the test on this dataset). The results obtained for the feature completion test are very similar to those of the row completion test. Again, we see that the number of feature completions is either zero / in very low digits or fairly large, with over 90% correctly completed feature values on some datasets.

For the first token test depicted in Table 6, we count the number of times that the model correctly predicts the first token of the next row in the CSV file. The prompts for this test are the same as for the row completion test; the difference is that we don't assess whether the entire row is correctly completed but only consider the first token. The first token test is especially interesting for the Adult dataset because we know from the construction of the dataset that the rows are ordered at random. In this case, a model that has never seen the dataset cannot be better at completing the first token than the mode. Interestingly, we see that the completion rate of GPT-4 is exactly at the rate of the mode.

### D.3    Additional Test Descriptions

Here we provide additional details on the different tests.

#### D.3.1    Feature Names

The feature names test asks the LLM to complete the names of the features of a dataset, given the name of the first feature. The test uses our general prompt structure with few-shot examples, given in Figure 14. In all cases, the result of this test was unambiguous. The model would either list the names of the different features in the same format as in the CSV file of the dataset, or it would offer clearly made-up feature names.

#### D.3.2    Feature Values

The feature values test asks the LLM to provide observations from the dataset that have valid feature values (most importantly, valid string values for categorical features). This test uses the prompt structure used in Section 5 in the main paper to draw samples from the dataset. We then manually check whether the formatting of the sampled feature values is the same as the formatting in the training data.

#### D.3.3    Feature Distribution

We ask the model to provide samples from the dataset at temperature 0.7. We then compare the model of the sample feature values with the mode of the feature in the data. This test is also motivated by the empirical observation that GPT-3.5 and GPT-4 model the modal feature values of many publicly available datasets surprisingly well, even if they rarely provide a good model of the finer details of the feature distribution.

#### D.3.4    Conditional Distribution

We draw samples at temperature 0.7 and compare the Pearson correlation coefficients of the samples with the Pearson correlation coefficients in the original dataset (compare Figure 16).

#### D.3.5    Header Test

For the header test, we split the dataset at random positions in rows 2, 4, 6, and 8 and ask the model to complete the dataset from the given position. We condition the model on this task with few-shot examples and consider the best completion. The test succeeds if the model

completes at least the next row in the dataset. This was unambiguous on all datasets (the model completed many rows or not even a single row). The prompt is depicted in Figure 13.

## E    Dataset Formats

Here, we describe the different dataset formats in more detail.

**Original.** We present the feature names and values as they occur in the CSV file of the dataset. In the system prompt, we tell the model about the origin of the data (*"You help to make predictions on the Titanic dataset from Kaggle"*).

**Perturbed.** Starting from the original data, we perturb continuous and discrete variables by randomly perturbing individual digits. We perturb the data so that we (1) change at least a single digit in all non-zero values in the data and (2) perturb with a relative error of approximately 1%. This means that the perturbation depends on a variable's magnitude and number of significant digits. The perturbations do not affect the format of the variables in the data, meaning that the perturbed data still "looks like" the original data. However, the numerical values are slightly different.

Because the goal of the perturbed version is to make it harder for the LLM to "recognize" the data, unique identifiers are given special treatment: All unique identifiers are changed to values that do not occur in the original dataset. We note that applying meaningful perturbations requires background knowledge (we only perturb variables that can be slightly changed without changing their meaning in the underlying problem).

For perturbed, we use the same system prompt as for original.

**Task.** Starting from the perturbed data, we re-format it. Numerical values are formatted to two decimal places by rounding and adding small amounts of noise. Categorical variables are re-coded ("0" becomes "False" and "United-States" becomes "USA") and features re-named ("BMI" becomes "Body Mass Index"). We also remove any hits on the original dataset from the system prompt ("You help to make predictions on the UCI Wine dataset, predicting the type of a wine from its features." becomes "You help to predict the type of a wine from its features.").

While we re-format the values under the task, we do not significantly change the values in the data. However, because the task builds on perturbed, the features still take slightly different values than when we had re-formatted the original dataset.

**Statistical.** Starting from the task data, we encode categorical variables into numeric values. All numeric features are standardized to zero mean and constant variance. The statistical transform was chosen to transform the appearance of the data without changing the underlying statistics.

Table 9 and Table 10 confirm that the different dataset transformations do not change the underlying statistical classification problem in any significant way.

## F    Replication with Llama 3.1 70B and Gemma 2 27B

In this Section, we replicate the experiments from Section 4 in the main paper using Llama 3.1 70B and Gemma 2 27B (Dubey et al., 2024; Riviere et al., 2024). We do not perform any changes to the setup of the experiment, that is we send the same prompts to the open-weight models that were previously send to GPT-3.5 and GPT-4 (including the ordering of the few-shot examples). We use https://www.together.ai/ as our inference engine, which means that the experiments can be performed with the same code that was used to send the prompts to the OpenAI API. The model endpoints are Meta-Llama-3.1-70B-Instruct-Turbo and google/gemma-2-27b-it. Code is available at https://github.com/interpretml/LLM-Tabular-Memorization-Checker.

While the experiment in this Section can be seen as a replication of the experiment in Section 4 in the main paper, **there is an important difference between the experiment detailed in**

| Panel A | Titanic | Adult | Diabetes | Wine | Iris |
|---|---|---|---|---|---|
| Original | $0.81_{.01}$ | $0.77_{.01}$ | $0.69_{.02}$ | $0.92_{.02}$ | $0.97_{.01}$ |
| Perturbed | $0.79_{.01}$ | $0.78_{.01}$ | $0.69_{.02}$ | $0.91_{.02}$ | $0.93_{.02}$ |
| Task | $0.78_{.01}$ | $0.72_{.01}$ | $0.70_{.02}$ | $0.87_{.03}$ | $0.94_{.02}$ |
| Statistical | $0.63_{.02}$ | $0.72_{.01}$ | $0.66_{.02}$ | $0.85_{.03}$ | $0.91_{.02}$ |
| LR / GBT | 0.79 / 0.80 | 0.86 / 0.87 | 0.78 / 0.75 | 0.98 / 0.96 | 0.97 / 0.95 |

| Panel B | S. Titanic | ACS Income | ICU | FICO | ACS Travel |
|---|---|---|---|---|---|
| Original | $0.69_{.01}$ | $0.78_{.01}$ | $0.69_{.05}$ | $0.69_{.01}$ | $0.62_{.02}$ |
| Perturbed | $0.69_{.01}$ | $0.78_{.01}$ | $0.71_{.05}$ | $0.68_{.01}$ | $0.62_{.02}$ |
| Task | $0.71_{.01}$ | $0.77_{.01}$ | $0.71_{.05}$ | $0.63_{.02}$ | $0.61_{.02}$ |
| Statistical | $0.66_{.01}$ | $0.59_{.02}$ | $0.57_{.05}$ | $0.61_{.02}$ | $0.49_{.02}$ |
| LR / GBT | 0.78 / 0.78 | 0.80 / 0.80 | 0.76 / 0.66 | 0.70 / 0.69 | 0.64 / 0.67 |

Table 7: Few-shot learning performance of LLama 3.1 70B across different tabular datasets. This table replicates Table 4 in the main paper.

| Panel A | Titanic | Adult | Diabetes | Wine | Iris |
|---|---|---|---|---|---|
| Original | $0.81_{.01}$ | $0.82_{.01}$ | $0.70_{.02}$ | $0.93_{.02}$ | $0.99_{.01}$ |
| Perturbed | $0.80_{.01}$ | $0.82_{.01}$ | $0.71_{.02}$ | $0.92_{.02}$ | $0.92_{.02}$ |
| Task | $0.79_{.01}$ | $0.78_{.01}$ | $0.72_{.02}$ | $0.90_{.02}$ | $0.92_{.02}$ |
| Statistical | $0.64_{.02}$ | $0.73_{.01}$ | $0.67_{.02}$ | $0.88_{.02}$ | $0.88_{.03}$ |
| LR / GBT | 0.79 / 0.80 | 0.86 / 0.87 | 0.78 / 0.75 | 0.98 / 0.96 | 0.97 / 0.95 |

| Panel B | S. Titanic | ACS Income | ICU | FICO | ACS Travel |
|---|---|---|---|---|---|
| Original | $0.68_{.01}$ | $0.77_{.01}$ | $0.73_{.04}$ | $0.64_{.02}$ | $0.63_{.02}$ |
| Perturbed | $0.68_{.01}$ | $0.77_{.01}$ | $0.70_{.05}$ | $0.64_{.02}$ | $0.64_{.02}$ |
| Task | $0.64_{.02}$ | $0.76_{.01}$ | $0.75_{.04}$ | $0.63_{.02}$ | $0.59_{.02}$ |
| Statistical | $0.65_{.02}$ | $0.59_{.02}$ | $0.54_{.05}$ | $0.54_{.02}$ | $0.54_{.02}$ |
| LR / GBT | 0.78 / 0.78 | 0.80 / 0.80 | 0.76 / 0.66 | 0.70 / 0.69 | 0.64 / 0.67 |

Table 8: Few-shot learning performance of Gemma 2 across different tabular datasets. This table replicates Table 4 in the main paper.

**this Section and the experiment in the main paper.** Namely, **the datasets that are "novel" for the versions of GPT-3.5 and GPT-4 considered in Section 4 might have been included in the training data of the open-weight models.** Moreover, as discussed in Section 3 in the main paper, the patterns of dataset memorization are different for the open-weight LLMs: Unlike GPT-3.5 and GPT-4, the open-weight LLMs do not seem to have memorized entire datasets, with the exception of the Iris dataset. Naturally, these two points are very important for the interpretation of the results, since we might not necessarily expect the same results if a dataset was included in the pre-training data, or if it was not memorized by the model.

With these caveats in mind, let us consider the few-shot learning results for Llama 3.1 70B and Gemma 2 27B. The results for Llama 3.1 are depicted in Table 7, and the results for Gemma 2 are depicted in Table 8. Figure 11 summarizes the results in both tables.

Interestingly, the results for Llama 3.1 70B are very similar to the results obtained with GPT-3.5 (compare Figure 11 and Figure 2 in the main paper). This is especially true on the popular tabular datasets, where both models show a similar decaying pattern across dataset transformations and also similar absolute performance (this is depicted in Subfigure (a) of

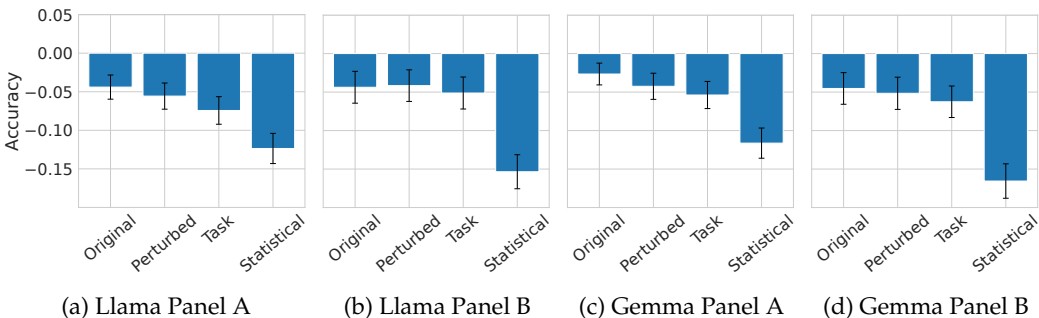

| (a) Llama Panel A | (b) Llama Panel B | (c) Gemma Panel A | (d) Gemma Panel B |

Figure 11: Few-shot learning performance of Llama 3.1 70B and Gemma 2 27B across the datasets depicted in Panel A and Panel B of Table 7 and Table 8, respectively. This figure replicates Figure 2 in the main paper. Note that we have replaced "memorized" and "novel" in the description of the Figure with "Panel A" and "Panel B". This is because the results of the memorization tests for Llama 3.1 70B and Gemma 2 27B indicate that there is less memorization than for GPT-3.5 and GPT-4.

Figure 11). On the datasets that are novel for GPT-3.5, both models show a similar pattern across dataset transformations, but Llama 3.1 70B performs, on average, better.

The results for Gemma 2 across the popular tabular datasets are similar to the results in the main paper, that is we see the characteristic decaying pattern across dataset transformations (this is depicted in Subfigure (c) of Figure 11). Gemma 2 depicts a slightly decaying pattern across dataset transformations also for the datasets that are novel for GPT-3.5 and GPT-4, even if this pattern is less pronounced and not statistically significant (this is depicted in Subfigure (d) of Figure 11). This result might be due to random fluctuations, or it might be that some of the datasets are included in the pre-training data of the model. More research would be needed to understand this in detail.

At a high level, it is interesting to see that **the results obtained with Llama 3.1 70 B and Gemma 2 27B are overall similar to the results obtained with GPT-3.5 and GPT-4**, even if some of the assumptions around the training data cutoff and memorization don't necessarily hold in the same way. Of course, this might be because the popular tabular datasets are still more likely to be included in the pre-training data than some of the novel datasets, even if the novel datasets were, strictly speaking, released before the cutoff of the training data.

## G   Prompts

Here we provide the example of a prompt used for few-shot learning on the OpenML Diabetes dataset in the original format.

---

**Few-shot learning with tabular data**

**System:**   You are a helpful statistician, data scientist, and board-certified medical doctor with expertise in treating diabetes.  You help to make predictions on the OpenML diabetes dataset.  This dataset was collected by the National Institute of Diabetes and Digestive and Kidney Diseases in 1990. Your task is to predict whether a patient tested positive or negative for diabetes. The user provides you with data on different individuals. You respond with '1' if your assessment is that the patient tested positive for diabetes and with '0' if your assessment is that the patient tested negative for diabetes. Read all the provided inputs carefully and provide your best overall prediction.

**User:** IF Pregnancies = 5.0, Glucose = 136.0, BloodPressure = 82.0, SkinThickness = 0.0, Insulin = 0.0, BMI = 0.0, DiabetesPedigreeFunction = 0.64, Age = 69.0, THEN
**Assistant:** 0

*[more few-shot examples]*

**User:** IF Pregnancies = 5.0, Glucose = 115.0, BloodPressure = 76.0, SkinThickness = 0.0, Insulin = 0.0, BMI = 31.2, DiabetesPedigreeFunction = 0.343, Age = 44.0, THEN
**Assistant:** 1

**User:** IF Pregnancies = 6.0, Glucose = 148.0, BloodPressure = 72.0, SkinThickness = 35.0, Insulin = 0.0, BMI = 33.6, DiabetesPedigreeFunction = 0.627, Age = 50.0, THEN

**Model Response:** 1

---

Figure 12: Few-shot learning with tabular data. The figure depicts few-shot learning on the original version of the OpenML diabetes dataset.

**Header test for memorization**

**System:** You are an autocomplete bot for tabular datasets. You will be prompted with parts of a tabular dataset. Your task is to complete the dataset.

**User:** sepal_length,sepal_width,petal_length,petal_width,species
5.1,3.5,1.4,0.2,Iris-setosa
4.9,3,1.4,0.2,Iris-setosa
4.7,3.2,1.3,0.2,Iris-setosa
[...]
4.8,3,1.4,0.1,Iris-setosa
4.3,3,1.1,0.1,Iris-setosa
5.8,4,1.2,0.2,Iris-s
**Assistant:** etosa
5.7,4.4,1.5,0.4,Iris-setosa
5.4,3.9,1.3,0.4,Iris-setosa
5.1,3.5,1.4,0.3,Iris-setosa
[...]

**User:** longitude,latitude,housing_median_age,[...],ocean_proximity
-122.23,37.88,41.0,880.0,129.0,322.0,126.0,8.3252,452600.0,NEAR BAY
-122.22,37.86,21.0,7099.0,1106.0,2401.0,1138.0,8.3014,358500.0,NEAR BAY
-122.24,37.85,52.0,1467.0,190.0,496.0,177.0,7.2574,352100.0,NEAR BAY
-122.25,37.85,52.0,1274.0,235.0,558.0,219.0,5.6431,341300.0,NEAR BAY
-122.25,37.85,52.0,1627.0,280.0,565.0,259.0,3.
**Assistant:** 8462,342200.0,NEAR BAY
-122.25,37.85,52.0,919.0,213.0,413.0,193.0,4.0368,269700.0,NEAR BAY
-122.25,37.84,52.0,2535.0,489.0,1094.0,514.0,3.6591,299200.0,NEAR BAY
-122.25,37.84,52.0,3104.0,687.0,1157.0,647.0,3.12,241400.0,NEAR BAY
-122.26,37.84,42.0,2555.0,665.0,1206.0,595.0,2.0804,226700.0,NEAR BAY
-122.25,37.84,52.0,3549.0,707.0,1551.0,714.0,3.6912,261100.0,NEAR BAY
-122.26,37.85,52.0,2202.0,434.0,910.0,402.0,3.2031,281500.0,NEAR BAY
-122.26,37.85,52.0,3503.0,752.0,1504.0,734.0,3.2705,241800.0,N

**User:** Age,WorkClass,fnlwgt,Education,[...],Income
39, State-gov,77516, Bachelors,13, Never-married, [...],0,40, United-States, <=50K
50, Self-emp-not-inc,83311, Bachelors,13, Married-civ-spouse, [...],0,13, United-States, <=50K
38, Private,215646, HS-grad,9, Divorced, Handlers-cleane

**Model Response:** rs, Not-in-family, White, Male,0,0,40, United-States, <=50K
53, Private,234721, 11th,7, Married-civ-spouse, [...],0,40, United-States, <=50K
28, Private,338409, Bachelors,13, Married-civ-spouse, [...],0,40, Cuba, <=50K
37, Private,284582, Masters,14, Married-civ-spouse, [...],0,40, United-States, <=50K
49, Private,160187, 9th,5, Married-spouse-absent, [...],0,16, Jamaica, <=50K
52, Self-emp-not-inc,209642, HS-grad,9, Married-civ-spouse, [...],0,45, United-States, >50K
[...]

Figure 13: The header test for memorization.

---

**Feature names test**

**System:** You are an expert assistant for tabular datasets. Your task is to list the names of the features of different datasets. The user provides a description of the dataset and some of the feature names. You then provide the names of the remaining features.

**User:** Dataset: IRIS. Feature Names: sepal_length
**Assistant:** sepal_width, petal_length, petal_width, species

**User:** Dataset: adult. Feature Names: Age
**Assistant:** WorkClass, fnlwgt, Education, EducationNum, MaritalStatus, Occupation, Relationship, Race, Gender, CapitalGain, CapitalLoss, HoursPerWeek, NativeCountry, Income

**User:** Dataset: titanic-train. Feature Names: PassengerId
**Assistant:** Survived, Pclass, Name, Sex, Age, SibSp, Parch, Ticket, Fare, Cabin, Embarked

**User:** Dataset: uci-wine. Feature Names: target
**Assistant:** alcohol, malic_acid, ash, alcalinity_of_ash, magnesium, total_phenols, flavanoids, nonflavanoid_phenols, proanthocyanins, color_intensity, hue, od280_od315_of_diluted_wines, proline

**User:** Dataset: california-housing. Feature Names: longitude
**Assistant:** latitude, housing_median_age, total_rooms, total_bedrooms, population, households, median_income, median_house_value, ocean_proximity

**User:** Dataset: fico. Feature Names: RiskPerformance

**Model Response:** ExternalRiskEstimate, MSinceOldestTradeOpen, MSinceMostRecentTradeOpen, AverageMInFile, NumSatisfactoryTrades, NumTrades60Ever2DerogPubRec, NumTrades90Ever2DerogPubRec, PercentTradesNeverDelq, MSinceMostRecentDelq, MaxDelq2PublicRecLast12M, MaxDelqEver, NumTotalTrades, NumTradesOpeninLast12M, PercentInstallTrades, MSinceMostRecentInqexcl7days, NumInqLast6M, NumInqLast6Mexcl7days, NetFractionRevolvingBurden, NetFractionInstallBurden, NumRevolvingTradesWBalance, NumInstallTradesWBalance, NumBank2NatlTradesWHighUtilization, PercentTradesWBalance

Figure 14: The feature names test.

---

**Sampling**

**System:** You are an expert assistant for tabular datasets. You provide random samples from different datasets. The user provides the name of the dataset and the names of the features. You then provide a sample from the dataset.

**User:** Dataset: IRIS
Feature Names: sepal_length, sepal_width, petal_length, petal_width, species
**Assistant:** sepal_length = 5.1, sepal_width = 3.5, petal_length = 1.4, petal_width = 0.3, species = Iris-setosa

**User:** Dataset: adult
Feature Names: Age, WorkClass, fnlwgt, Education, EducationNum, MaritalStatus, Occupation, Relationship, Race, Gender, CapitalGain, CapitalLoss, HoursPerWeek, NativeCountry, Income
Feature Names: Age, WorkClass, fnlwgt, Education, EducationNum, MaritalStatus, Occupation, Relationship, Race, Gender, CapitalGain, CapitalLoss, HoursPerWeek, NativeCountry, Income
**Assistant:** Age = 43, WorkClass = Self-emp-inc, fnlwgt = 196945, Education = HS-grad, EducationNum = 9, MaritalStatus = Married-civ-spouse, Occupation = Other-service, Relationship = Husband, Race = Asian-Pac-Islander, Gender = Male, CapitalGain = 0, CapitalLoss = 0, HoursPerWeek = 78, NativeCountry = Thailand, Income = $<= 50K$

**User:** Dataset: openml-diabetes
Feature Names: Pregnancies, Glucose, BloodPressure, SkinThickness, Insulin, BMI, DiabetesPedigreeFunction, Age, Outcome
**Assistant:** Pregnancies = 1, Glucose = 95, BloodPressure = 74, SkinThickness = 21, Insulin = 73, BMI = 25.9, DiabetesPedigreeFunction = 0.673, Age = 36, Outcome = 0

**User:** Dataset: uci-wine
Feature Names: target, alcohol, malic_acid, ash, alcalinity_of_ash, magnesium, total_phenols, flavanoids, nonflavanoid_phenols, proanthocyanins, color_intensity, hue, od280_od315_of_diluted_wines, proline
**Assistant:** target = 1, alcohol = 13.24, malic_acid = 2.59, ash = 2.87, alcalinity_of_ash = 21.0, magnesium = 118, total_phenols = 2.8, flavanoids = 2.69, nonflavanoid_phenols = 0.39, proanthocyanins = 1.82, color_intensity = 4.32, hue = 1.04, od280_od315_of_diluted_wines = 2.93, proline = 735

**User:** Dataset: titanic-train
Feature Names: PassengerId, Survived, Pclass, Name, Sex, Age, SibSp, Parch, Ticket, Fare, Cabin, Embarked
**Assistant:** PassengerId = 746, Survived = 0, Pclass = 3, Name = Abbott, Mr. Rossmore Edward, Sex = male, Age = 16.0, SibSp = 1, Parch = 1, Ticket = C.A. 2673, Fare = 20.25, Cabin = NaN, Embarked = S

**User:** Dataset: california-housing
Feature Names: longitude, latitude, housing_median_age, total_rooms, total_bedrooms, population, households, median_income, median_house_value, ocean_proximity

**Model Response:** longitude = -122.12, latitude = 37.68, housing_median_age = 45.0, total_rooms = 2179.0, total_bedrooms = 401.0, population = 1159.0, households = 399.0, median_income = 3.4839, median_house_value = 180600.0, ocean_proximity = NEAR BAY

Figure 15: Sampling from the California Housing dataset.

## H   Additional Tables and Figures

| Panel A.   | Titanic | Adult | Diabetes | Wine | Iris |
|------------|---------|-------|----------|------|------|
| Original   | 0.79 | 0.86 | 0.78 | 0.98 | 0.97 |
| Perturbed  | 0.79 | 0.86 | 0.78 | 0.98 | 0.97 |
| Task       | 0.78 | 0.86 | 0.77 | 0.97 | 0.98 |
| Statistical| 0.79 | 0.86 | 0.77 | 0.98 | 0.97 |

| Panel B.   | S. Titanic | ACS Income | ICU | FICO | ACS Travel |
|------------|------------|------------|-----|------|------------|
| Original   | 0.78 | 0.80 | 0.77 | 0.70 | 0.64 |
| Perturbed  | 0.78 | 0.80 | 0.75 | 0.69 | 0.64 |
| Task       | 0.78 | 0.80 | 0.76 | 0.70 | 0.64 |
| Statistical| 0.77 | 0.80 | 0.75 | 0.70 | 0.64 |

Table 9: Test accuracy of Logistic Regression on the different dataset versions. This table addresses the question: "Do the perturbations and transformations that we perform with the data change the underlying classification problem significantly?". We see that this is not the case. In particular, the results with the original data are the same as with the statistically transformed data (up to perhaps a single percentage point).

| Panel A.   | Titanic | Adult | Diabetes | Wine | Iris |
|------------|---------|-------|----------|------|------|
| Original   | 0.80 | 0.88 | 0.75 | 0.96 | 0.95 |
| Perturbed  | 0.79 | 0.87 | 0.76 | 0.95 | 0.95 |
| Task       | 0.80 | 0.87 | 0.75 | 0.96 | 0.93 |
| Statistical| 0.79 | 0.86 | 0.76 | 0.97 | 0.95 |

| Panel B.   | S. Titanic | ACS Income | ICU | FICO | ACS Travel |
|------------|------------|------------|-----|------|------------|
| Original   | 0.78 | 0.80 | 0.67 | 0.68 | 0.67 |
| Perturbed  | 0.77 | 0.80 | 0.63 | 0.69 | 0.68 |
| Task       | 0.77 | 0.80 | 0.66 | 0.69 | 0.67 |
| Statistical| 0.78 | 0.80 | 0.66 | 0.68 | 0.67 |

Table 10: Test accuracy of a Gradient Boosted Tree on the different dataset versions. This table addresses the question: "Do the perturbations and transformations that we perform with the data change the underlying classification problem significantly?". We see that this is not the case. In particular, the results with the original data are the same as with the statistically transformed data.

| | California Housing | |
|---|---|---|
| | GPT-3.5 | GPT-4 |
| Number of samples copied from the training data | 0.1% | 10.4% |
| Average number of features shared with the closest match in the training data | 3.1 / 10 | 4.4 / 10 |
| Fraction of individual feature values that also occur in the training data | 99.53% | 99.64% |

Table 11: Summary statistics of samples from California Housing.

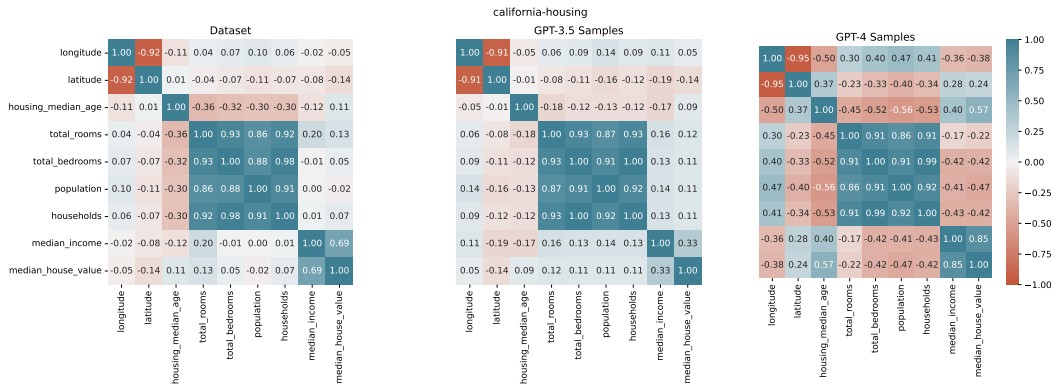

Figure 16: Pearson correlation coefficients of samples from California Housing.

