# OpenReview forum: "Elephants Never Forget: Memorization and Learning of Tabular Data in Large Language Models"
_colmweb.org/COLM/2024/Conference — COLM_

### Official Review · Reviewer_odGz · 2024-05-10

**Rating:** 7
**Confidence:** 2
**Ethics Flag:** 1

**Summary:**

Disclaimer: I am an NLP researcher/computational linguist and do not feel qualified to review this paper. It applies LLMs to tasks that are pretty much outside of NLP.

This paper discusses experiments that use LLMs as "statistical predictors" or classifiers on tabular data, e.g. classification on the famous Iris datasets. It shows that GPT3.5 and GPT4 have memorized a couple of such popular tabular datasets. It describes experiments that perturb these datasets in systematic ways and reports the resulting performance in LLMs.

**Questions To Authors:**

See weaknesses above -- more details on how the memorization tests have been evaluated would be very welcome.

**Reasons To Accept:**

Assuming that researchers now really start to seriously use LLMs as classifiers on tabular datasets, it is an important contribution to show that LLMs have memorized some common datasets in this domain (for this community).

I found the idea of the systematic perturbation of tabular dataset convincing, as this seems to be a straightforward way to show that LLMs have limited ability to act as statistical predictors (which is not surprising to me at all, but still, it might be important to show this).

**Reasons To Reject:**

The memorization tests for the tabular datasets seem somewhat simple and are described superficially. The "tests" prompt the LLM with certain elements of the table and record whether the LLM regenerates parts of the tabular data. The results of these tests are presented in a table that just shows some red crosses/green checkmarks for +/- memorized and it is unclear how these crosses and checkmarks have been computed. The supplement gives more detailed numbers, but I could not determine why certain tests/datasets have checkmarks and others do not. For instance, why does the "Adult" dataset have a checkmark for the header test and red crosses for the others? (In supplement D, it looks like the GPTs cannot regenerate any of the elements in this dataset)

It is unclear how exactly the tests have been conducted and evaluated (how many runs, how many rows, features etc.). For instance, which portions of regenerated rows are considered as indications of memorization?

The results of memorization tests do not seem entirely conclusive, for instance in the "Adult" dataset. Is the dataset memorized or not? What exactly are the end results of the test battery for each dataset? As one of the main contributions of this paper is to show memorization of tabular data, I would have expected more experiments, substance, and systematic validation on this.

This is not really a weakness of the paper per se, but I cannot stop myself from writing this here: given everything we know about how LLMs are built and trained, it is pretty clear that they are *not* reliable statistical predictors. I believe that using an LLM for this task is an "unintended" and maybe even an "unreasonable" use. I would expect at least some discussion and reflection on this in the paper. Since LLMs are now widely used by lay users, unintended uses like this one do come with ethical concerns and should be approached with some care.

Post-rebuttal: The discussion with the authors brought up some interesting points and clarified most of my concerns ... I have adjusted my score accordingly.

---

> ### Author Rebuttal · Authors · 2024-05-30
>
> We thank the reviewer for taking the time to review our paper. We appreciate that the reviewer is candid about the fact that they don’t feel entirely qualified to review the paper.
>
> We commit to include additional details about how the memorization tests are conducted and evaluated. In particular, we will specify how the pass/fail checkmarks in Table 2 relate to the quantitative test results in Supplement Table 4. We will also include qualitative test results for the header test.
>
> The reviewer asks why the "Adult" dataset has a checkmark for the header test but red crosses for the other tests. The header test measures the memorization of the dataset's initial rows, whereas the other tests measure the memorization of random rows. While we do not have access to the training data of the LLMs, the likely reason is that the initial rows of the “Adult” dataset are replicated on the internet more often than random rows, which leads to memorization (for example, due to the common practice of printing the initial rows of a dataset in Jupyter Notebooks). In line with the results of our tests, it turns out that a Google search for the initial rows of the “Adult” dataset leads to search results, whereas there are no results for random rows.
>
> *"given everything we know about how LLMs are built and trained, it is pretty clear that they are not reliable statistical predictors"*: There is quite some evidence that LLMs can act as statistical predictors (Garg et al. 2022 https://arxiv.org/abs/2208.01066). It has even been shown that in-context learning with a transformer can mimic least squares regression (Oswald et al. 2023 https://arxiv.org/abs/2212.07677). GPT-3.5 and GPT-4 are trained not only on natural language but also on structured data (like the tabular datasets reported in the paper), so we don’t think it entirely implausible that they acquire the ability to act as statistical predictors.
>
> We again thank the reviewer for the review. We would be happy to engage in discussion and would appreciate it if the reviewer would decide to increase their score.

---

> > ### Comment · Reviewer_odGz · 2024-06-03
> > **Response to rebuttal**
> >
> > Thanks for the interesting response. The reference to Oswald et al. 2023 would have helped me lot, to understand the general background of this work.
> >
> > Regarding the memorization tests, I am still not entirely happy. Could you give some more details as to how you defined "memorization" quantitatively? Or is Table 2 obtained from "eyeballing"/manually checking the test battery results?
> >
> > Your explanation on the adult dataset suggests that memorization is not a binary thing, but that LLMs sometimes seem to memorize particular parts of a dataset.  This is an interesting observation, but it is not discussed in the paper if I am not mistaken. What would be the consequences of "partial memorization"? Do you think that it differs from "full memorization"?

---

> > ### Author Response · Authors · 2024-06-05
> > **Response to comment**
> >
> > Thanks for your response!
> >
> > Although the memorization tests lead to quantitative results, such as the number of rows that were correctly completed, we don't define memorization quantitatively.
> >
> > Our notion of memorization follows the approach outlined in a series of papers on memorization in LLMs, most importantly Carlini et al. 2019 https://arxiv.org/abs/1802.08232 and Carlini et al. 2021 https://arxiv.org/abs/2012.07805.
> >
> > Following this literature, we say a text is memorized if a model can consistently generate it.
> >
> > Of course, this makes sense only for texts that contain sufficient entropy so that the probability of generating the text by chance or prediction is close to zero.
> >
> > In our context of tabular datasets, we rely on the fact that these datasets contain random variables: It is impossible to consistently reproduce the realizations of random variables unless the values of the random variables have been seen before (i.e., during pre-training).
> >
> > For example, the OpenML Diabetes dataset contains an individual's glucose level, blood pressure, and BMI, as well as other measurements that are at least in part random. Now, if an LLM can consistently generate entire rows of this dataset, this is evidence of memorization. We note that this approach has been lightly formalized in the two papers referenced above, where the authors use the term "canary."
> >
> > In judging memorization, one has to consider the completion rate of the LLM and the amount of entropy in the dataset. To give another example, the Iris dataset contains many rows that are near-duplicates. This means that an LLM might also achieve a non-zero row completion rate by chance or prediction. Because of this complication, we manually judge the results in Table 2, given the results in Supplement Table 4, and the structure of the dataset.
> >
> > Thankfully for our approach, it turns out that  GPT-3.5 and GPT-4 (as well as other LLMs) have the remarkable property of regurgitating quite long and complex strings from their training data. For example, GPT-4 is able to generate *the entire Iris dataset*, consisting of 151 rows, given only the first couple of rows as prompt. Thus, while there are few cases of ambiguity (marked as question marks in Table 2), the evidence of memorization is usually fairly clear (and conversely, there are the cases where the model gets the dataset's content completely wrong).
> >
> > To better illustrate the nature of our memorization tests, we have created additional Figures depicting the qualitative test results: https://github.com/colm365/code/blob/main/memorization_tests.pdf. The figures illustrate the Leveshtein string distance between the model response and the actual content in the dataset. For the different tests, we depict two cases, one where there is evidence of memorization and one where there is no evidence of memorization.
> >
> > We would like to highlight that the anonymous code repository contains a Python package that can perform all the memorization tests (and the package can also print the prompts).
> >
> > We again thank the reviewer for engaging in the discussion and would be happy to answer any further questions.

---

> > > ### Comment · Reviewer_odGz · 2024-06-06
> > > **Response to comment**
> > >
> > > Thanks, this helped me to understand what is going on in the memorization tests. I hope that some of these comments and visualizations can find their way into the paper! I have adjusted my score, but I will keep my confidence low as the paper really is outside of my area of expertise.

---

### Official Review · Reviewer_sEZf · 2024-05-12

**Rating:** 8
**Confidence:** 4
**Ethics Flag:** 1

**Summary:**

Paper summary: This paper investigates how seeing a dataset during pretraining affects the model's performance on the dataset.  They consider datasets that are stored in tabular form, specifically CSV format, as this is a very common format for storing datasets.  They develop novel exposure tests for this type of data, and find that LLMs perform much better on datasets they have seen during pretraining.  They also investigate the in-context learning abilities of LLMs, and find that without fine-tuning, their in-context learning abilities to be limited.

Review summary: The quality, clarity, originality and significance of this work is quite high.  It is a very good fit for COLM.

**Reasons To Accept:**

1) Exposure of LLMs to training data is important issue that needs to be addressed.

2) The contributions of this paper are as follows:
2.1) They develop novel techniques for detecting training data contamination for tabular datasets.
2.2) For tabular datasets, they find that "The striking difference in performance between the different dataset formats on the memorized datasets, which is completely absent on the novel datasets, strongly suggests that memorization leads to invalid performance estimates of few-shot learning." (page 6, bottom).  This is novel for tabular datasets.  However, this is not novel for datasets in general, as prior work has found this to be the case as well "Task Contamination: Language Models May Not Be Few-Shot Anymore" Li & Flanigan 2023 https://arxiv.org/abs/2312.16337).

**Reasons To Reject:**

They call this type of contamination "training data contamination".  In prior work, this has been called "task contamination" (see "Task Contamination: Language Models May Not Be Few-Shot Anymore" Li & Flanigan 2023 https://arxiv.org/abs/2312.16337).  While it's ok that they have used a different term for this, they need to cite this previous work.  Additionally, the above paper somewhat reduces the novelty claims in introduction.

---

> ### Author Rebuttal · Authors · 2024-05-30
>
> We thank the reviewer for taking the time to review our paper. We appreciate that the reviewer finds our paper a very good fit for COLM.
>
> Thanks for pointing us to Li & Flanigan 2023. We did not know about this paper. We will expand the related works section and include this paper.
>
> Insofar as the term “(training) data contamination” is concerned, we note that it is somewhat established in the literature (see, e.g., Magar & Schwarz 2023 https://arxiv.org/pdf/2203.08242 or Jiagn et al. 2024 https://arxiv.org/pdf/2401.06059). We will reference the “task contamination” terminology and note that this is an emerging research area.
>
> We again thank the reviewer for appreciating our paper.

---

### Official Review · Reviewer_ZJeM · 2024-05-12

**Rating:** 5
**Confidence:** 3
**Ethics Flag:** 1

**Summary:**

The paper introduces different techniques to evaluate if a language model has seen a specific tabular dataset during training. Second, the authors evaluate fe-shot performance of datasets which were seen during training and datasets that were published after the cut-off date of the LLM training data.  They use a varied set of datasets and perturbations on top of them to study the memorization and few-shot learning capabilities of GPT-3.5 and GPT4.

**Reasons To Accept:**

The paper presents a rich suite of tabular datasets and transformation strategies that can be applied on this type of data. These resources can be a valuable resource of evaluating LLM capabilities on non-linguistic data.

**Reasons To Reject:**

First, I consider that the evaluation only on GPT-3.5 and GPT-4 models is limited and it would be much more valuable to study the memorisation issues on a different set of LLMs and especially open-source models (e.g. Olmo). Also, this addition would help understanding if these datasets are present in the training data and their frequency and not having to use heuristics for predicting if a dataset was present during pertaining.

I also think that the paper lacks a deeper explanation on the evidences of memorisation of the tabular datasets from the training data.

---

> ### Author Rebuttal · Authors · 2024-05-30
>
> We thank the reviewer for taking the time to review our paper. We appreciate that the reviewer believes the paper can be a valuable resource for evaluating LLM capabilities.
>
> In our paper, we study the problem of evaluating closed-source LLMs where we do not have access to the training data or the model weights. We show that training data contamination can lead to overfitting during few-shot learning, a common evaluation setup in the literature for these models.
>
> The reviewer notes that it would be interesting to study open-source LLMs. We **agree** with the reviewer. Indeed, this setting has several advantages. Most importantly, it can be known how often a particular text was seen during pre-training.
>
> However, we **disagree** with the reviewer that it would be *"much more valuable to study the memorisation issues on a different set of LLMs."* Instead, we believe that it is valuable to study training data contamination for both open-source and closed-source LLMs.
>
> To see why it is valuable to study training data contamination for closed-source LLMs that are highly capable, consider an LLM that overfits on a particular dataset because it has memorized the dataset, but not learned to perform the underlying task. As we increase the overall training data and parameters of the LLM, the model might still memorize the dataset but also learn to perform the underlying task. This means that for highly capable LLMs, we might observe memorization without overfitting. Because of this empirical posibility, it's valuable to study training data contamination for the most capable (closed-source) LLMs.
>
> We highlight that the literature on memorization already demonstrates the value of studying both open-source and closed-source LLMs  (for example, Carlini et al. 2021 https://arxiv.org/abs/2012.07805,  Biderman et al. 2023 https://arxiv.org/abs/2304.11158, Nasr et al. 2023 https://arxiv.org/abs/2311.17035).
>
> We release all of our code as a Python package, which allows replication studies with different datasets and LLMs.
>
> What is more, our memorization tests for closed-source models are novel. With the extended page limit of 10 pages, we commit to expanding the discussion of these tests.
>
> We again thank the reviewer for the review. We would be happy to engage in discussion and would appreciate it if the reviewer would decide to increase their score.

---

> ### Comment · Area_Chair_kgLP · 2024-06-04
>
> Dear Reviewer, could you please check if the author's answers are helpful and if there is any follow up question that you have?

---

### Official Review · Reviewer_MP4s · 2024-05-20

**Rating:** 7
**Confidence:** 3
**Ethics Flag:** 1

**Summary:**

The paper addresses the issue of data contamination and memorization in LLMs, particularly focusing on tabular data. It introduces various techniques to test whether a language model has seen specific datasets during training. The study finds that GPT-3.5 and GPT-4 can memorize popular tabular datasets verbatim and that these models perform better on datasets they have seen before. This memorization can lead to overfitting. However, the models show non-trivial performance on new, unseen datasets and are somewhat robust to data transformations. The paper concludes that careful testing using the developed exposure tests is essential to determine if a dataset has been seen during pre-training.

Main Findings:
1. Data Contamination: The study reveals that LLMs can memorize large portions of tabular datasets they were trained on, impacting their performance evaluation.
2. Few-Shot Learning Performance: The models perform better on datasets they have seen before. Adding small amounts of noise or reformatting reduces this performance, indicating memorization and data contamination.
3. Performance on Unseen Data: Despite memorization, GPT-3.5 and GPT-4 show significant performance on novel datasets, meaning they are not solely relying on previously seen data.
4. Comparison with Baseline: When compared to traditional statistical learning algorithms, GPT-3.5 and GPT-4 still perform competitively even on unseen datasets, particularly in few-shot learning scenarios.
5. Robustness to Transformations: Even though the models seem robust to certain data transformations, careful evaluation is required to ensure integrity in performance assessments.

Without a doubt, I find this paper valuable for the community. My only concern is the limited set of datasets. Most of them are standard in ML,DL learning which makes them obvious candidates to be memorized. There are several less popular datasets before 2021 that could be used to have a broader spectrum of results.

**Reasons To Accept:**

The paper addresses critical concerns around data contamination and memorization in LLMs, which are increasingly important given the widespread use of these models in various applications.
While much research on LLMs focuses on natural language, this paper's focus on tabular data is both novel and relevant, extending the understanding of LLMs to structured data.
IMO The introduction of various techniques to determine whether an LLM has seen a specific dataset during training is a valuable contribution. These methods can be used by other researchers to ensure the integrity of their evaluations.
The paper employs multiple methods to test memorization, including adding noise and reformatting data.

**Reasons To Reject:**

It has more than 9 pages, the appendices also make it particularly long.
A limited set of datasets. Most of them are standard in ML,DL learning which makes them obvious candidates to be memorized.
The related work section looks somewhat lacking; it's evident that the authors sacrificed this section to give space to others.
The use of the word 'elephant' in the title is somewhat strange; the analogy seems excessive to me."

---

> ### Author Rebuttal · Authors · 2024-05-30
>
> We thank the reviewer for taking the time to review our paper. We appreciate that the reviewer finds the paper valuable for the community.
>
> We agree that many of the datasets in the paper are standard in ML. We chose these datasets specifically because they seemed likely candidates to be memorized. Note, however, that this is not the case for all the datasets in our paper — we even constructed a novel tabular dataset specifically for this study (the ICU dataset).
>
> The reviewer observes that it would be interesting to study datasets that might have been in the training data of the LLM, but which are less common than the datasets considered in our study. We agree that this would be interesting. However, this would be challenging in our setup, where we don't know the training data. This is because either there is evidence of memorization, which likely means that the dataset was seen very often during training (hence, "very common"), or there is no evidence of memorization, which means that we are essentially unable to say whether the dataset was part of the training corpus or not (because membership inference attacks for sota LLMs are challenging, see e.g. Duan et al. https://arxiv.org/abs/2402.07841).
>
> Future work, however, might tackle this question about less common datasets.  In this regard, note that we release all of our code as a Python package so that other researchers who might have access to the training data of some LLMs can perform replication studies with different datasets and LLMs.
>
> With the extended page limit of 10 pages, we commit to expanding the related work section.
>
> We again thank the reviewer for the review.

---

> > ### Comment · Reviewer_MP4s · 2024-05-31
> >
> > Thank you for the responses! I agree that knowing the pre-training data is impossible, but clues can be used to determine what is common and what is not. For example, the number of views/downloads (available on the UCI repository), the publication date, the number of papers citing it, the number of Google search results for a query, etc. You can include it in your future work!

---

### Decision · Program_Chairs · 2024-07-10

**Decision:**

Accept

**Comment:**

This submission investigates the critical issue of data contamination and memorization in large language models (LLMs), specifically focusing on tabular data. The paper introduces various techniques to assess whether an LLM has seen a particular tabular dataset during training. The authors find that GPT-3.5 and GPT-4 have memorized many popular tabular datasets verbatim, leading to better performance on datasets seen during training, indicating overfitting due to memorization. However, the models also demonstrate non-trivial performance on novel datasets and robustness to certain data transformations.

The reviewers acknowledge the paper's value in addressing an important concern regarding data contamination and memorization in LLMs. The focus on tabular data is novel and extends the understanding of LLMs to structured data. The introduction of various techniques to determine whether an LLM has seen a specific dataset during training is a valuable contribution that can be used by other researchers to ensure the integrity of their evaluations.

Reviewer MP4s raises a minor concern about the limited set of datasets used, suggesting that the inclusion of less popular datasets could provide a broader spectrum of results. Reviewer ZJeMS suggests evaluating the memorization issues on a different set of LLMs, especially open-source models, and providing a deeper explanation of the evidence for memorization of tabular datasets from the training data. Reviewer sEZf notes that while the findings on the effects of memorization on few-shot learning are novel for tabular datasets, similar observations have been made for datasets in general in prior work, which should be cited.

Overall, the reviewers agree that the paper addresses a critical issue, provides valuable contributions, and should be accepted for publication. The outcome is a clear accept. Please address all suggestions of the reviewers for the final version.  A personal request, in line with reviewer ZJeMS is the application of this study to a open source LLMs (open datasets included). This would compare the findings in this paper with a transparent fine-tuning setup.

[comment from the PCs] Please follow up on the AC's request regarding open-weight/source models. This is important for research.

[At least one review was discounted during the decision process due to quality]